# Mitochondrial dysfunction alters early endosome trafficking via microtubule reorganization

Anjali Vishwakarma[1,2], Lilia Chihki[1,2], Kiran Todkar[1,2], Mathieu Ouellet[1,2], Marc Germain[1,2]

**Mitochondria are essential for bioenergetics and cellular processes including cell differentiation and immunity; alterations in these processes cause a wide range of muscular and neurological pathologies. Although these pathologies have traditionally been associated with ATP deficits, mitochondrial dysfunction also leads to reactive oxygen species (ROS) generation, inflammation, and alterations in the function of other organelles. Although the negative impact of mitochondrial dysfunction on lysosomal activity is established, the relationship between mitochondria and the rest of the endocytic compartment remains poorly understood. Here, we show that inhibiting mitochondrial activity through genetic and chemical approaches causes early endosome (EE) perinuclear aggregation and impairs cargo delivery to lysosomes. This impairment is due to ROS-mediated alterations in microtubule architecture and centrosome dynamics. Antioxidants can rescue these EE defects, underlying the pivotal role of mitochondria in maintaining cellular activities through ROS regulation of microtubule networks. Our findings highlight the significance of mitochondria beyond ATP production, emphasizing their critical involvement in endocytic trafficking and cellular homeostasis. These insights emphasize mitochondria's critical involvement in cellular activities and suggest novel targets for therapies to mitigate the effects of mitochondrial dysfunction.**

## Introduction

Mitochondria are key bioenergetic organelles that play crucial roles not only in metabolite generation and exchange, but also in a range of cellular processes including cell differentiation and immunity (1, 2). As a consequence, alterations in mitochondrial functions cause a wide range of muscular and neurological pathologies (3). Although these pathologies were historically thought to result from ATP deficits, it is now clear that other mechanisms are also involved, including enhanced inflammation and the generation of reactive oxygen species (ROS) (4, 5). In fact, the ATP-independent roles of mitochondria are crucial for the control of cellular differentiation during development, inflammation, and cancer (6).

At the cellular level, altered mitochondrial structure and function are linked to defects in other organelles. For example, loss of mitochondrial function leads to defects in lysosomal activity (7, 8). As lysosomes are the main degradative organelles, this affects protein turnover and causes the accumulation of aggregated material within the affected cells. Lysosomes receive their material to degrade through autophagy, which is known to be affected by mitochondrial dysfunction, and after endocytosis of extracellular material (9, 10). In the endocytic pathway, extracellular material first accumulates in early endosomes (EEs) that sort this material towards recycling to the cell surface or target them to late endosomes and lysosomes for degradation (11). The sorting and transfer of material to recycling and late endosomes require the action of small GTPases of the Rab family. Rabs coordinate all the steps required for cargo sorting and transfer to other endocytic organelles, including the transport along microtubules that allow fusion between vesicles and transfer to lysosomes (12).

Although alterations in mitochondrial activity impact lysosomes, the consequences on other components of the endocytic pathway remain poorly understood. Here, we show that mitochondrial dysfunction disrupts EE distribution and cargo trafficking, revealing a previously unrecognized link between mitochondrial function and endocytic transport mechanisms. Specifically, the presence of genetic mutations in mitochondrial proteins or the chemical inhibition of the electron transport chain results in the perinuclear aggregation of EEs and impairs their ability to deliver cargo to lysosomes. These alterations in EE distribution are caused by changes in microtubule architecture and centrosome dynamics, which then facilitates EE transport and aggregation in the perinuclear region of the cell. We found that this

[1]Groupe de Recherche en Signalisation Cellulaire et Département de Biologie Médicale, Université du Québec à Trois-Rivières, Trois-Rivières, Canada [2]Centre d'Excellence en Recherche sur les Maladies Orphelines - Fondation Courtois, Université du Québec à Montréal, Montréal, Canada

Correspondence: marc.germain1@uqtr.ca
Kiran Todkar's present address is Department of Cellular and Molecular Medicine, University of Ottawa, Ottawa, Canada
Mathieu Ouellet's present address is Department of Electrical and Systems Engineering, School of Engineering and Applied Science, University of Pennsylvania, Philadelphia, PA, USA

is mediated by ROS, which affect microtubules and centrosome organization and thus EE distribution and function. Consequently, antioxidants rescue EE defects present in cells with mitochondrial dysfunction. Overall, our results demonstrate that ROS generated by defective mitochondria actively disrupt endocytic trafficking through the reorganization of microtubule networks, underscoring the critical role of mitochondria in the maintenance of cellular activities.

# Results

### Mitochondrial dysfunction causes early endosome aggregation

Mitochondrial dysfunction impairs lysosomal activity, which is accompanied by the presence of enlarged and vacuolated lysosomes and late endosomes (7, 8, 13). We thus determined whether this extended to the rest of the endocytic compartment. For this, we first used MEFs in which the essential mitochondrial protein OPA1 is deleted as a model for mitochondrial dysfunction (8, 14). As we previously reported (8), these cells contained large vacuolated structures marked with the lysosomal marker LAMP1 (Fig 1A and B). As early endosomes (EEs) are required to deliver extracellular material to lysosomes, we determined whether they become vacuolated similar to lysosomes by staining cells with an antibody against the EE marker Rab5. However, in contrast to LAMP1-positive lysosomes, Rab5-positive EEs did not appear vacuolated in OPA1 KO MEFs (Fig 1A and B). Nevertheless, there was a clear shift in EE distribution in OPA1 KO cells. Whereas EEs were equally dispersed throughout in control cells, we observed perinuclear clustering of EEs in the mutant cells, which was seen as large aggregates of Rab5-positive structures (Fig 1A, C, and D). On the other hand, LAMP1-positive lysosomes did not accumulate in the perinuclear region (Fig 1E), although their distribution was somewhat altered (with overall decreased amounts in the middle of the cytoplasm relative to the perinuclear and plasma membrane regions, Fig S1A). Rab11-positive recycling endosomes were present in the perinuclear region irrespective of the genotype (Figs 1F and S1B and C), as previously reported for WT cells (15).

To confirm that the alterations we observed in EEs are caused by the impairment of mitochondrial function, not some other defect associated with mutant fibroblasts, we chemically inhibited subunits of the electron transport chain (ETC) in WT MEFs. Similar to OPA1 KO MEFs, rotenone (complex I inhibitor) or antimycin A (AA, complex III inhibitor) caused the perinuclear aggregation of Rab5-positive EEs (Fig 1G). AA treatment also caused EE perinuclear aggregation in HeLa cells (Fig 1H). Interestingly, a similar aggregation of rab5-positive EEs was observed in primary human fibroblasts mutant for the mitochondrial fission protein DRP1 (Fig 1I). To confirm that the changes are the consequence of altered EEs, not simply a change in Rab5 distribution, we measured EE distribution using EEA1 as a second EE marker. Similar to Rab5-positive EEs, we found that EEA1-positive early endosomes aggregated in the perinuclear region of OPA1 KO MEFs, WT MEFs

treated with ETC inhibitors, and DRP1 mutant fibroblasts (Figs 1J–L and S1D), further indicating that EEs are altered in these cells.

### EE aggregation in cells with mitochondrial dysfunction is associated with impaired EE cargo trafficking

The altered EE distribution in cells with mitochondrial dysfunction prompted us to address its consequences on EE function. EEs serve as sorting organelles to transfer cargo to late endosomes/lysosomes for degradation, or recycling endosomes for material returning to the cell surface (11). We first tracked the transfer of EE cargo towards lysosomes using fluorescently labelled dextran. Cells were first pulsed for 5 min with dextran, after which dextran uptake was evaluated by immunofluorescence and quantified by measuring the number of dextran-positive vesicles present in each cell. As cells with mitochondrial dysfunction showed similar uptake of dextran than control cells (Fig 2A), we then performed a chase experiment where the fate of the dextran is followed over time after the initial pulse. We investigated the trafficking of dextran in these endosomes by measuring the association of dextran with Rab5 and the lysosomal marker LAMP1 by confocal microscopy. As expected, dextran colocalized with Rab5-positive EEs immediately after the initial pulse, and gradually transferred to LAMP1-positive late endosomes/lysosomes over time in control cells (Fig 2B and C). We also confirmed this manual evaluation of colocalization by measuring Pearson's correlation coefficients in the same images. Although the overall low amount of colocalization between dextran and Rab5 caused low Pearson's coefficient values, there was a clear increase in Pearson's coefficient between dextran and LAMP1 over time, confirming our colocalization data (Fig S2A and B).

Having validated the assay, we then measured dextran distribution in the endosomes of OPA1 KO MEFs. In contrast to WT cells, dextran colocalized with Rab5-positive EEs to a much lower extent in OPA1 KO MEFs. Consistent with this, dextran transferred to LAMP1-positive lysosomes at a lower rate in OPA1 KO MEFs (Figs 2B and C and S2A and B), demonstrating a defect in cargo delivery from the EEs to lysosomes in these cells. Similar results were obtained when WT MEFs were treated with AA (Figs 2B and C and S2A and B).

We then examined the recycling function of EEs by tracking the delivery of cargo from EEs to recycling endosomes using fluorescently labelled transferrin (Tf). Tf binds to the Tf receptor at the cell surface, liberates its associated iron within EEs, and is then recycled to the cell surface in recycling endosomes (16). As with dextran, an initial pulse of Tf led to the endocytosis of a similar amount of Tf in control, OPA1 KO, and AA-treated MEFs, as judged by the similar number of Tf-positive vesicles present in these cells (Fig 2D). In control cells, the total number of transferrin spots then dropped, as expected because it is recycled to the cell surface (Fig 2D). Tf foci also decreased in OPA1 KO cells and WT cells treated with AA, although at a somewhat reduced rate (% left at 30 min: WT 12% ± 4%; KO 29% ± 8%, $P$ = 0.022; AA 31% ± 2%, $P$ = 0.015). Interestingly, the colocalization with Rab5-positive EEs was increased in cells with mitochondrial dysfunction (Figs 2E and S2C). In addition, the colocalization between Tf and the recycling endosome marker Rab11 was increased at the beginning of the chase time but not at later time points (Figs 2F and S2D). Overall, these results indicate

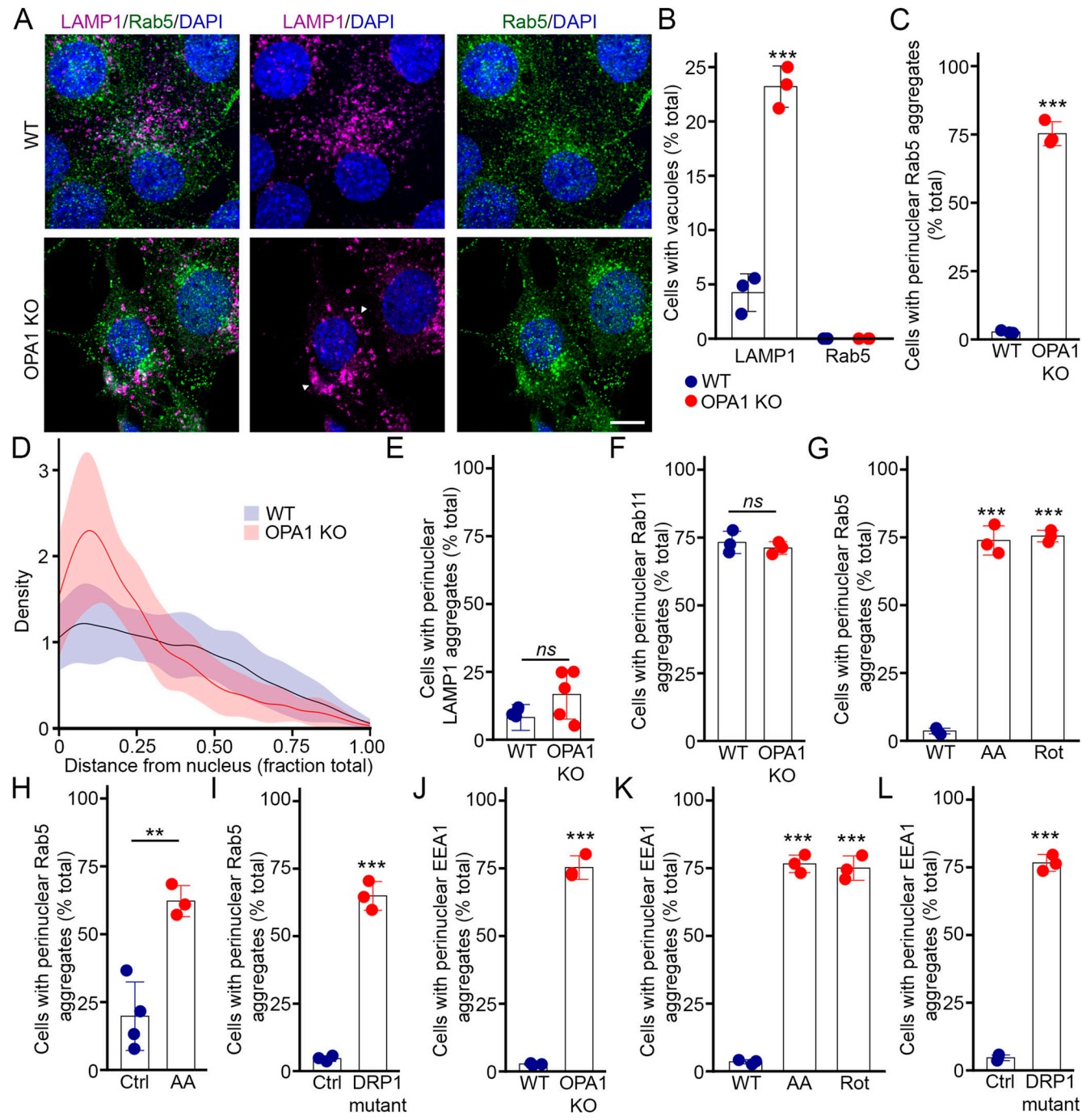

**Figure 1. Mitochondrial dysfunction alters the distribution of early endosomes.**
**(A)** Representative confocal images of WT and OPA1 KO MEFs stained for the lysosomal marker LAMP1 (magenta) and the EE marker Rab5 (green), along with DAPI to mark nuclei (blue). Arrowheads denote enlarged vacuolated lysosomes. Scale bar, 10 $\mu$m. **(A, B)** Quantification of WT and OPA1 KO cells with LAMP1-positive and Rab5-positive vacuoles from images in (A). Each point represents an independent experiment. Bars show the average ± SD for three experiments per condition. ***$P$ < 0.001, two-sided $t$ test. **(A, C, D, E, F)** Quantification of perinuclear clustering of the endosomal markers Rab5 (C), LAMP1 (E), and Rab11 (F) in WT and OPA1 KO MEFs from confocal images as in (A). Each point represents an independent experiment. Bars show the average ± SD for three experiments per condition (five for LAMP1). ***$P$ < 0.001, two-sided $t$ test. ns, not significant. **(D)** Density of Rab5-positive vesicles relative to their localization was also measured for Rab5 from the same images. The data show the quantification of 30 cells per condition in three independent experiments ± SD. **(G)** Quantification of perinuclear clustering of Rab5 in WT MEFs treated with the complex III inhibitor antimycin A (AA) or the complex I inhibitor rotenone (Rot). Each point represents an independent experiment. Bars show the average ± SD for three experiments per condition. ***$P$ < 0.001, one-way ANOVA. **(H)** Quantification of Rab5 perinuclear clustering in AA-treated HeLa cells. Each point represents an independent experiment. Bars show the average ± SD for three experiments per condition. **$P$ < 0.01, two-sided $t$ test. **(I)** Quantification of Rab5 perinuclear clustering in control and DRP1 mutant primary fibroblasts. Each point represents an independent experiment. Bars show the average ± SD for three experiments per condition. ***$P$ < 0.001, two-sided $t$ test. **(J, K, L)** Quantification of perinuclear clustering of the EE marker EEA1 in OPA1 KO MEFs (J), WT MEFs treated with AA or rotenone (K), and DRP1 mutant primary fibroblasts (L). Bars show the average ± SD for three experiments per condition. ***$P$ < 0.001, two-sided $t$ test, one-way ANOVA for (K).
Source data are available for this figure.

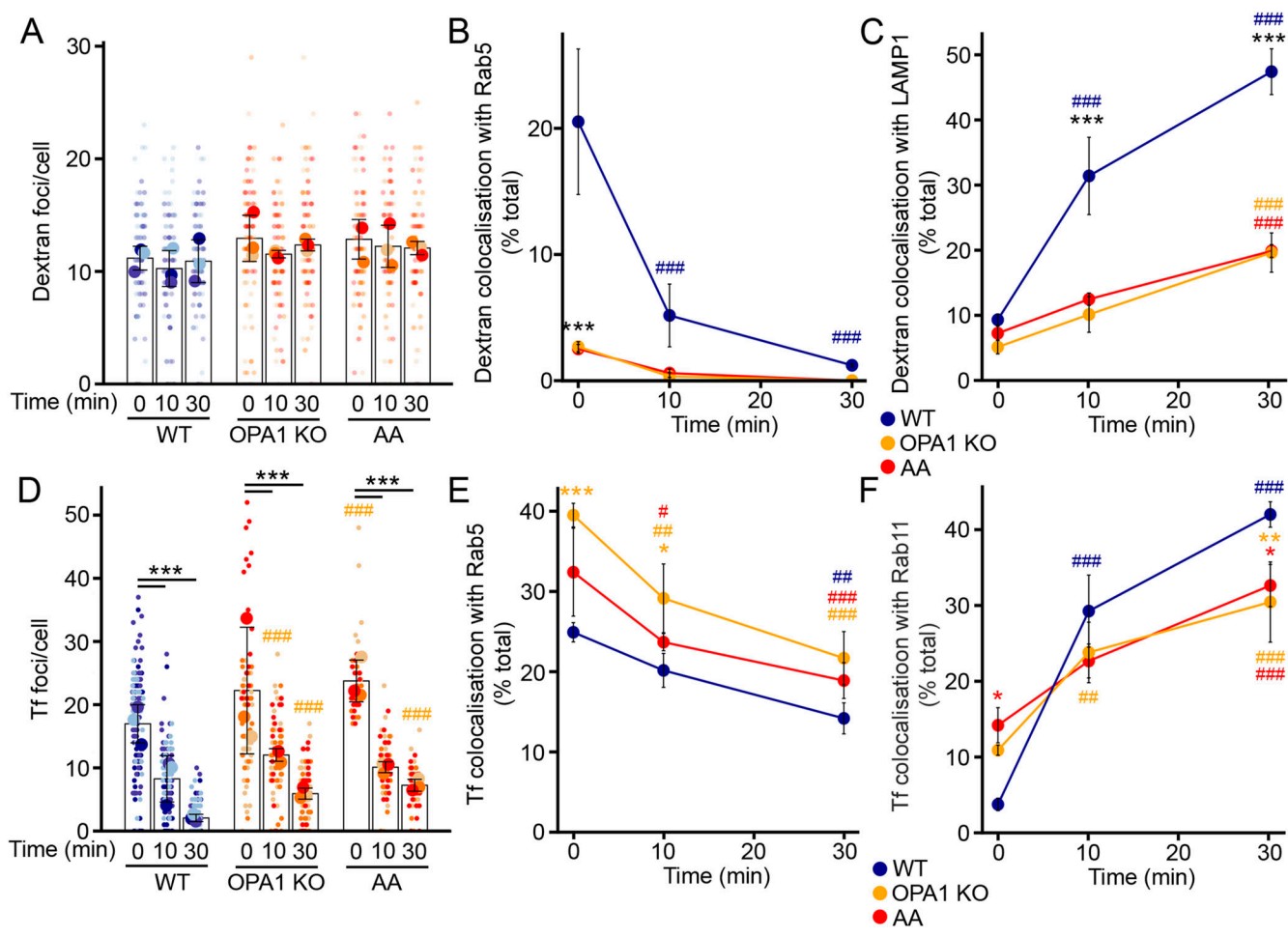

**Figure 2. Mitochondrial dysfunction impairs proper EE cargo trafficking.**
**(A, B, C)** Dextran trafficking in OPA1 KO MEFs and AA-treated WT MEFs. **(A, B, C)** Cells were pulsed with dextran for 5 min, then chased for the indicated times, and total dextran (A) and its colocalization with Rab5 (B) and LAMP1 (C) were quantified by immunofluorescence. Each point represents an independent experiment, with small points in (A) representing individual cells (number of cells: 97 [WT], 101 [KO], 92 [AA]). Bars show the average ± SD for three experiments per condition. ***P < 0.001 versus WT, ###P < 0.001 versus 0 min, two-way ANOVA. **(D, E, F)** Transferrin (Tf) trafficking in OPA1 KO MEFs and AA-treated WT MEFs. Cells were pulsed with Tf for 5 min, then chased for the indicated times, and total Tf (D) and its colocalization with Rab5 (E) and Rab11 (F) were quantified by immunofluorescence. Each point represents an independent experiment, with small points in (D) representing individual cells (number of cells: 118 [WT], 115 [KO], 62 [AA]). Bars show the average ± SD for three experiments per condition. ***P < 0.001 versus WT, ###P < 0.001 versus 0 min, two-way ANOVA. All stats are calculated from experiment averages, not cell averages. Source data are available for this figure.

that both lysosome delivery and endosomal recycling are affected by impaired mitochondrial activity, although in a distinct manner.

### EE clustering is driven by microtubule-dependent retrograde transport

Mitochondrial dysfunction induces perinuclear aggregation of EEs, resulting in altered EE spatial distribution that subsequently impacts cargo trafficking towards lysosomes. To elucidate the underlying cause, we initially determined the behaviour of EEs by live-cell imaging. WT and OPA1 KO cells were transfected with RFP-Rab5 and imaged over time using a confocal microscope. We then analysed the movement of RFP-Rab5-positive EEs in these cells by measuring the radial (towards/away from the nucleus) and angular (side-to-side) components of vesicle speed (Fig 3A). This revealed that although the angular velocity of the EEs was not

significantly affected by OPA1 deletion, their radial velocity was significantly increased (Fig 3B and C). Specifically, and consistent with the perinuclear aggregation of EEs we observed, mutant cells exhibited an augmented velocity of EEs directed towards the nucleus relative to control cells (Fig 3D). We then determined whether these alterations were selective for EEs by repeating the experiment in cells in which lysosomes were tagged using GFP-LAMP1. In contrast to Rab5-positive vesicles, the radial velocity of LAMP1-positive vesicles was not altered in OPA1 KO MEFs (Fig 3B and D), although a subset of cells showed increased angular velocity (Fig 3C). Altogether, these results indicate that EE movements are increased in OPA1 KO cells.

As EE transport occurs along microtubules, we next investigated the relationship between EE aggregation and microtubule transport in cells with mitochondrial dysfunction. For this, we first stained WT and OPA1 KO cells for Rab5, the microtubule protein

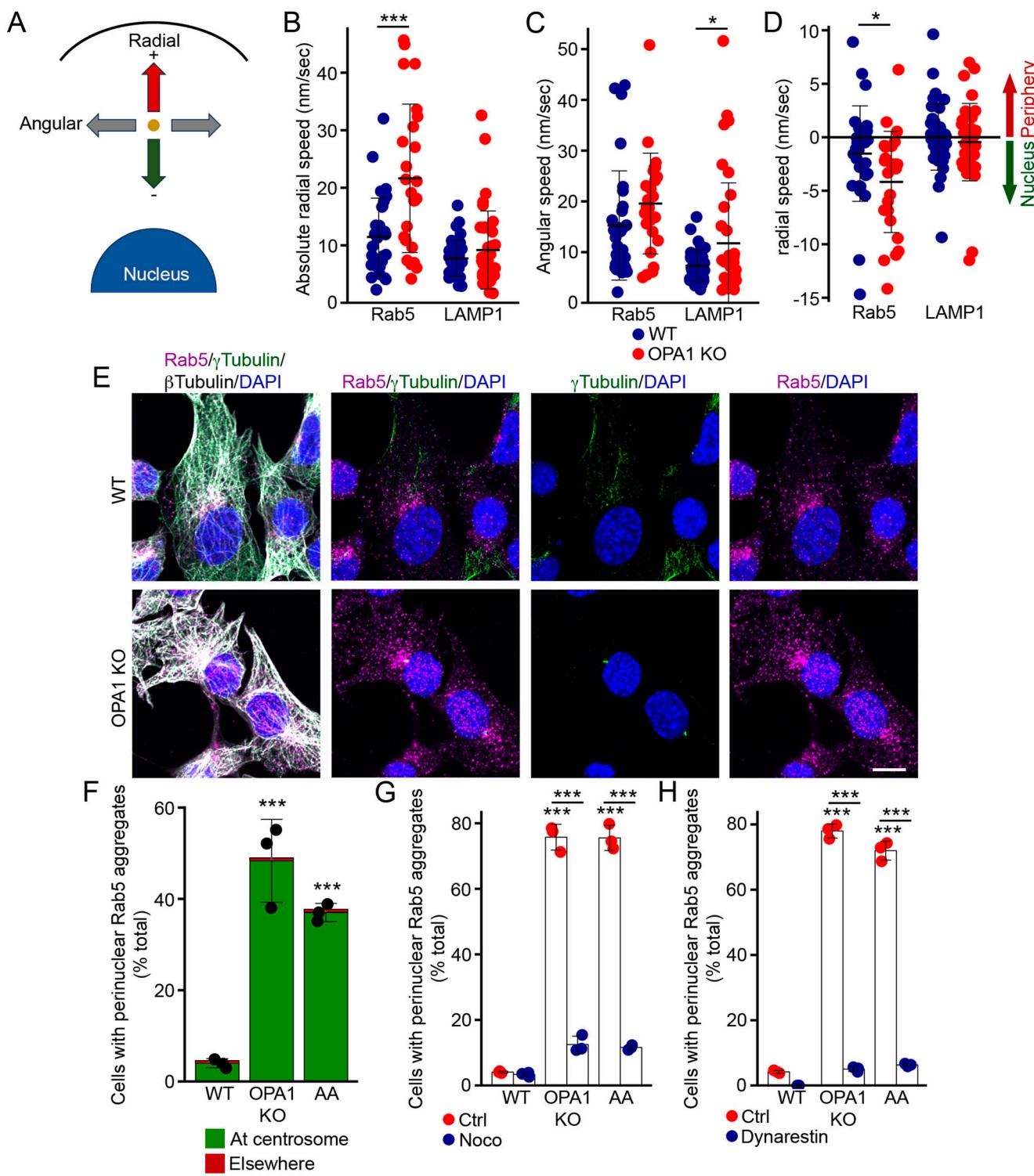

**Figure 3. Microtubules drive EE perinuclear clustering.**
**(A, B, C, D)** Quantification of EE and lysosome velocity in OPA1 KO MEFs transfected with either Rab5-RFP (EE) or LAMP1-GFP (lysosome). **(A)** Schematic representation of radial and angular velocity. **(B, C, D)** Absolute radial speed (independent of direction), (C) angular speed, (D) radial speed according to its direction (negative towards the nucleus, positive towards the cell membrane). Each point represents an individual endosome, with endosomes analysed in a minimum of four cells per condition in at least three independent experiments. Bars show the average ± SD. ***P < 0.001, *P < 0.05, two-sided t test. **(E, F)** Rab5-positive EEs cluster at centrosomes. **(E)** Representative confocal images of WT and OPA1 KO MEFs stained for the EE marker Rab5 (magenta), along with the microtubule marker ß-tubulin (white) and the centrosome marker γ-tubulin (green). Scale bar, 10 μm. **(F)** Quantification of images in (E). Each point represents an independent experiment. Bars show the average ± SD for three experiments per condition. ***P < 0.001, one-way ANOVA. **(G, H)** Inhibition of microtubule-dependent transport rescues Rab5-positive EE distribution in OPA1 KO MEFs. WT and OPA1 KO MEFs were treated with the microtubule inhibitor nocodazole (noco, (G)) or the dynein inhibitor dynarrestin (H) in the absence or the

ß-tubulin, and the centrosome marker γ-tubulin. In control cells, where EEs were dispersed throughout the cell, no association between EEs and centrosomes was observed (Fig 3E and F). Conversely, in mutant cells characterized by the aggregation of early endosomes in the perinuclear region, we identified the minus end of microtubules and centrosomes associating with the aggregated early endosomes (Fig 3E and F). This triple association substantiates the involvement of microtubules and centrosomes in the aggregation of early endosomes.

To investigate the causal relationship between microtubules and the altered distribution of early endosomes, we employed the microtubule-depolymerizing agent nocodazole as an experimental tool. A short pulse of nocodazole (15 min) had no impact on the distribution of Rab5-positive EEs in control cells (Fig 3G). In contrast, nocodazole treatment dispersed aggregated EEs from the perinuclear region in mutant cells (Fig 3G). A similar redistribution of Rab5 upon nocodazole treatment was observed in WT cells treated with AA (Fig 3G), consistent with microtubule-dependent transport being required for perinuclear EE aggregation. The motor protein dynein plays a key role in the retrograde transport of EE (17). Thus, we then determined the effect of the dynein inhibitor dynarrestin on Rab5 clustering in cells with mitochondrial dysfunction. Similar to the disruption of microtubules with nocodazole, dynarrestin treatment caused a redistribution of Rab5-positive EEs in OPA1 KO cells and in AA-treated WT cells (Fig 3H). Altogether, these results indicate that microtubule-based transport promotes perinuclear EE aggregation in our cellular models of mitochondrial dysfunction.

### Mitochondrial dysfunction leads to centrosome alterations

To determine the reason why microtubule-based transport specifically altered EE localization in cells with mitochondrial dysfunction, we first assessed potential alterations in microtubule organization. Most interphase cells contain one centrosome from where microtubules are organized, whereas dividing cells have duplicated their centrosome in prevision of chromosome segregation and cytokinesis (18). Consistent with this, almost all WT MEFs contained either one (57%) or two (41%) centrosomes (Fig 4A and B), as detected by immunofluorescence using the centrosomal marker γ-tubulin. In contrast, OPA1 KO cells and AA-treated WT cells displayed a decrease in cells with one centrosome, and a significant proportion of cells with more than two centrosomes (Fig 4A and B). In addition, although the two centrosomes were apart from each other in WT cells, as would be expected from cells preparing to divide, the distance between the two centrosomes decreased as the number of centrosomes increased in OPA1 KO and AA-treated cells (Fig 4C), suggesting that duplicated centrosomes fail to segregate in these cells.

In osteoclasts, centrosome clustering facilitates microtubule bundling, thereby enhancing microtubule-based transport (19). A similar bundling was present in OPA1 KO MEFs and AA-treated WT

MEFs, as seen by the presence of much thicker microtubule fibres in cells with closely apposed duplicated centrosomes, compared with control cells (Fig 4D and E). Importantly, the presence of these dense microtubules correlated with EE perinuclear aggregation (Figs 3E and 4E). We also assessed microtubule bundling using microtubule directionality, a measure based on the dispersion of the microtubule signal. In this assay, a decrease in the dispersion of the fluorescent signal of microtubules (i.e., bundling) is seen as a decrease in the SD of the signal, which is exactly what we observed in OPA1 KO cells and AA-treated cells (Fig 4F). Collectively, our results suggest that alterations in centrosomes lead to changes in the organization of microtubules, promoting rapid dynein-driven EE movement and their perinuclear aggregation.

To test this more directly, we manipulated centrosome numbers by overexpressing PLK4, a kinase regulating centrosome duplication that promotes the presence of supernumerary centrosomes when overexpressed (20, 21). The expression of GFP-tagged PLK4 resulted in the presence of either several centrosomes dispersed throughout the cell (69% of transfected cells; Dispersed [D], Fig S3A) or two closely juxtaposed centrosomes (31% of transfected cells; Cluster of 2 [C2], Fig S3A). Consistent with our results, cells with a cluster of two centrosomes, but not cells with dispersed centrosomes, had bundled microtubules (Fig S3B and C) and aggregated Rab5 vesicles (Fig 4G). These results thus support the idea that centrosome alterations drive the endosomal phenotype present in cells with impaired mitochondrial activity.

### Oxidative stress promotes centrosome duplication and EE aggregation

Mitochondrial dysfunction often leads to an increase in ROS (22, 23), and we previously showed that this was associated with the impaired lysosomal structure and function found in these cells (8). Consistent with this, OPA1 KO MEFs and AA-treated cells showed an increase in oxidized proteins as detected using oxyblots (Figs 5A and S4A). The level of oxidized proteins nevertheless remained low compared with the exposure of cells to $H_2O_2$, which causes a large and rapid increase in oxidative stress (Fig 5A).

Oxidative stress can also cause aberrant centrosome duplication, leading to the presence of supernumerary centrosomes and aberrant cell division (24). To directly test whether this leads to EE aggregation, we sought to induce a low but sustained level of exogenous ROS akin to what is found in our models of mitochondrial dysfunction. We thus added glucose oxidase (GO) to cell media, to generate a controlled amount of ROS in WT cells (8). GO caused a significant increase in the number of centrosomes, leading to the appearance of clustered centrosomes similar to OPA1 KO and AA-treated cells (Fig 5B–D). This was associated with an increase in the number of cells with bundled microtubules as quantified directly (Fig 5E) and by measuring microtubule directionality (Fig 5F). Consistent with ROS-driven microtubule alterations being responsible for Rab5 aggregation, exposure of WT cells to GO also caused the perinuclear aggregation of Rab5-

---

presence of AA as indicated. Each point represents an independent experiment. Bars show the average ± SD for three experiments per condition. ***$P < 0.001$, one-way ANOVA.
Source data are available for this figure.

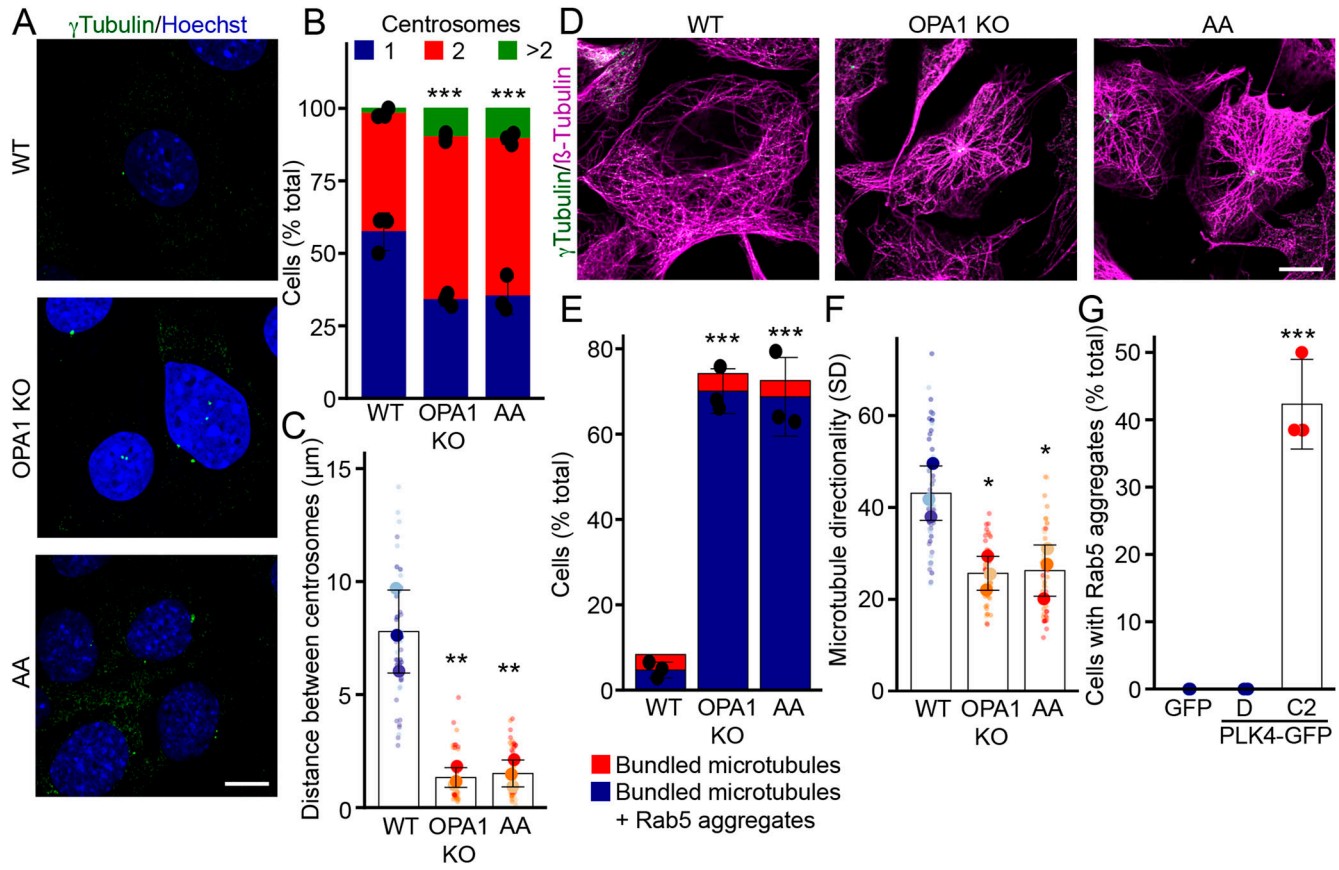

**Figure 4. Mitochondrial dysfunction leads to aberrant centrosome duplication.**
**(A)** Representative confocal images of WT (control and AA-treated) and OPA1 KO MEFs stained for the centrosome marker γ-tubulin (green) along with DAPI (blue) to mark nuclei. Scale bar, 10 μm. **(B)** Quantification of the number of centrosomes per cell from images in (A). Each point represents an independent experiment. Bars show the average ± SD for three experiments per condition. ***$P < 0.001$, two-way ANOVA. **(C)** Quantification of the distance between centrosomes from images in (A). Each point represents an independent experiment, with smaller points representing individual cells. Bars show the average ± SD for three experiments per condition. ***$P < 0.001$, one-way ANOVA. **(D)** Representative confocal images of WT (control and AA-treated) and OPA1 KO MEFs stained for the centrosome marker γ-tubulin (green) and the microtubule marker ß-tubulin (magenta). Scale bar, 10 μm. **(E)** Quantification of the correlation between the presence of bundled centrosomes and Rab5 aggregates in OPA1 KO and AA-treated cells. Each point represents an independent experiment. Bars show the average ± SD for three experiments per condition. ***$P < 0.001$, two-way ANOVA. **(F)** Microtubule bundling was also quantified by measuring the variation in microtubule directionality expressed as the SD. Each point represents an independent experiment, with smaller points representing individual cells. Bars show the average ± SD for three experiments per condition. *$P < 0.05$, one-way ANOVA. **(G)** PLK4 expression causes EE aggregation. WT MEFs were transfected with the indicated plasmids, and Rab5 aggregates were quantified by immunofluorescence. D refers to PLK4-transfected cells with dispersed centrosomes, whereas C2 refers to transfected cells with two clustered centrosomes (see Fig S3). Each point represents an independent experiment. Bars show the average ± SD for three experiments per condition. ***$P < 0.001$, one-way ANOVA. All stats are calculated from experiment averages, not cell averages.
Source data are available for this figure.

positive EEs (Fig 5G), similar to what we observed in our cellular models of mitochondrial dysfunction. Importantly, a single large burst of oxidative stress caused by exposing cells to $H_2O_2$ did not cause microtubule alterations or Rab5 aggregation (Fig S4B–E), consistent with a previous report (25). Overall, our data indicate that a small increase in ROS phenocopies the defects present in OPA1 KO MEFs, but that this is not the consequence of non-specific endosomal damage caused by high $H_2O_2$ levels.

To further demonstrate that ROS cause the centrosome alterations leading to EE clustering, we quenched ROS in our cellular models of mitochondrial dysfunction using antioxidants. Consistent with our GO data, N-acetyl cysteine (NAC) decreased the number of OPA1 KO and AA-treated cells that contained more than two centrosomes and increased the distance between centrosomes (Fig 5H and I, total

centrosome numbers in Fig S5A). Similarly, we observed a rescue of microtubule bundling and directionality in antioxidant-treated cells (Fig 5J and K). This rescue of microtubule structure caused the dispersion of EEs from the perinuclear region towards the entire cell (Fig 5L), supporting a key role for ROS in this process.

We then determined whether mitochondrial ROS were required for this process. For this, we first used the mitochondria-targeted antioxidant MitoQ. Similar to NAC, MitoQ rescued centrosomes, microtubules, and Rab5 distribution in both OPA1 KO and AA-treated cells (Fig 5H and K and S5A). Mitochondria generate ROS in the matrix and in the intermembrane space in the form of superoxide, which is then converted to $H_2O_2$ by superoxide dismutase. To further support the role of mitochondria-derived ROS in Rab5 aggregation and to determine whether $H_2O_2$ is specifically

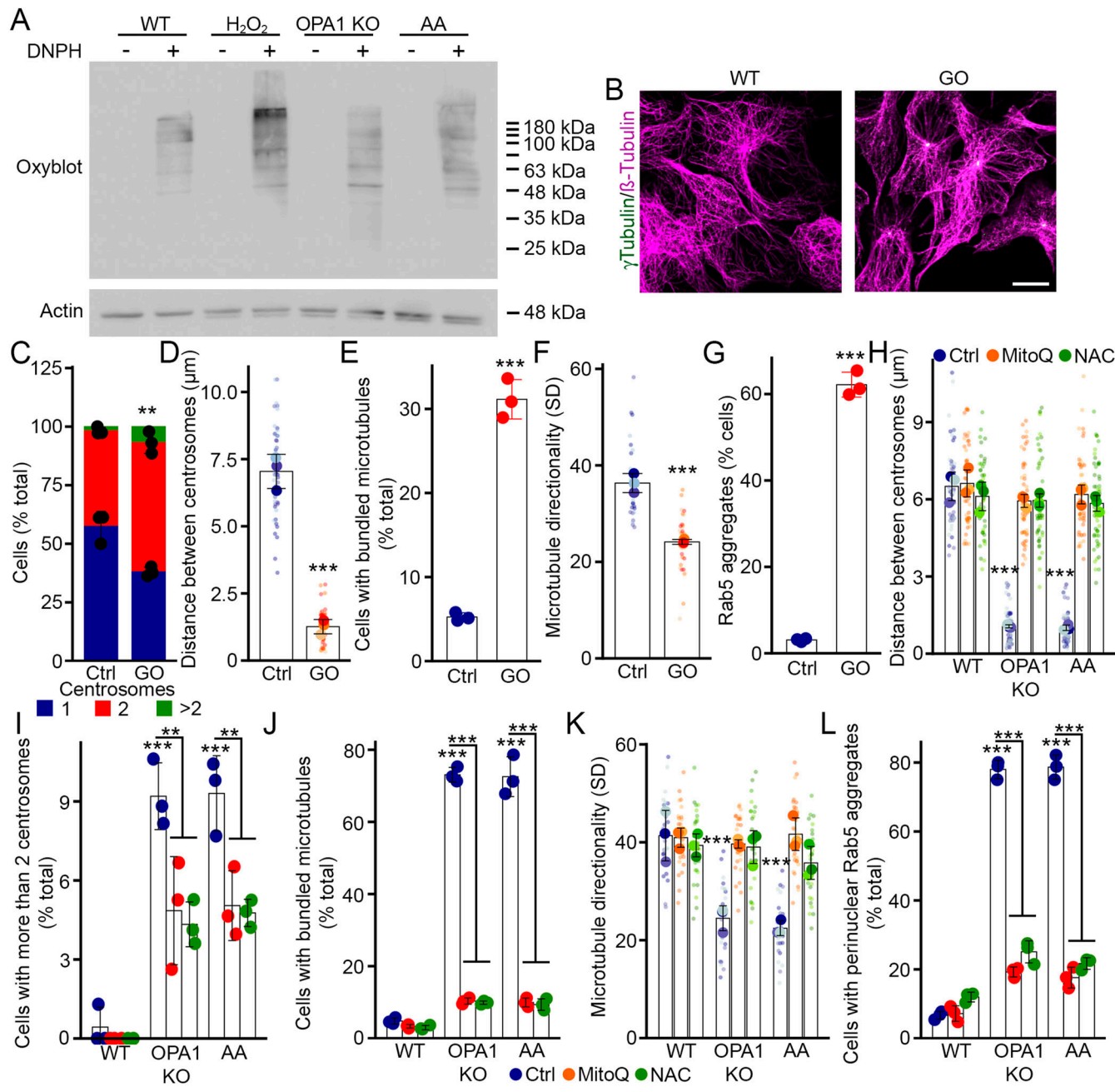

**Figure 5. Oxidative stress promotes centrosome alterations leading to EE perinuclear clustering.**
**(A)** Oxidative stress increases the levels of oxidized proteins. Oxyblot showing the amount of oxidized proteins in the indicated condition in the absence (DNPH −) or the presence (DNPH +) of the labelling chemical. Actin was used as a loading control. **(B, C, D, E, F, G)** Glucose oxidase (GO) promotes centrosome alterations and EE clustering. **(B)** Representative confocal images of control and GO-treated WT MEFs stained for the microtubule marker ß-tubulin (magenta) and the centrosome marker γ-tubulin (green). Scale bar, 10 μm. **(C, D, E, F)** Number of centrosomes (C), the distance between them (D), and the number of cells with bundled microtubules (E) and their directionality (F) were quantified from these images. **(G)** The presence of Rab5 aggregates was also quantified. Each point represents an independent experiment, with smaller points in (D, F) representing individual cells. Bars show the average ± SD for three experiments per condition. ***$P < 0.001$, **$P < 0.01$, two-sided $t$ test. **(H, I, J, K, L)** Antioxidants rescue both centrosome defects and EE aggregation. WT and OPA1 KO MEFs were treated as indicated. **(H, I, J, K)** Distance between centrosomes (H) and the number of cells with more than two centrosomes ((I, total distribution of centrosomes in Fig S5A) were quantified, as well as the number of cells with bundled microtubules (J) and their directionality (K). Rab5 aggregates were also quantified (L) from confocal images as in (B, C, D, E, F, G). Each point represents an independent experiment, with smaller points in (H, K) representing individual cells. Bars show the average ± SD for three experiments per condition. ***$P < 0.001$, **$P < 0.01$, two-way ANOVA. All stats are calculated from experiment averages, not cell averages.
Source data are available for this figure.

involved in this process, we expressed catalase specifically in the mitochondrial matrix using a previously described construct (26). In our hands, this construct targeted catalase to mitochondria in about 50% of the cells; the other transfected cells show high cytosolic expression (Fig S5B; catalase was detected using an antibody that also recognizes the peroxisome-localized endogenous protein as seen in the top panel). Consistent with $H_2O_2$ being the main ROS responsible for the EE phenotype, cytosolic catalase fully rescued Rab5 localization and microtubule structure (Fig S5C and D). Similarly, matrix-targeted catalase partially rescued EEs and microtubules (Fig S5C and D), further indicating that mitochondria-derived $H_2O_2$ promotes microtubule-dependent EE aggregation. The partial rescue in this context is likely because matrix-targeted catalase quenches matrix-generated $H_2O_2$ but not the $H_2O_2$ generated in the intermembrane space that can still escape to the cytosol. Collectively, our data demonstrate that oxidative stress–induced changes in microtubule organization and centrosomes contribute to the altered EE distribution.

To further define the link between ROS and microtubule organization, we then assessed the effect of centrosome amplification on ROS production. Although AA treatment led to an increase in ROS as measured using MitoSOX, PLK4 transfection did not (Fig S5E), supporting the idea that ROS act upstream of centrosome alterations. The kinase ROCK1 was previously shown to promote centrosomal amplification in response to ROS (27). Consistent with this, treatment of OPA1 KO cells with a ROCK1 inhibitor (Y-27632) rescued both microtubule structure and EE distribution (Fig S5F and G), suggesting that mitochondria-generated ROS alter EE distribution by causing ROCK1-dependent microtubule alterations.

## Oxidative stress causes a functional loss of cargo trafficking

As ROS promote EE aggregation, we then investigated the impact of oxidative stress on the trafficking capacity of EEs. We first exposed control cells to GO to induce an increase in ROS production. Consistent with our models of mitochondrial dysfunction, GO did not decrease the overall uptake of dextran (Fig 6A). However, GO reduced the colocalization between dextran and Rab5-positive EEs and reduced trafficking towards LAMP1-positive lysosomes (Fig 6B–E), similar to cells with mitochondrial dysfunction (Fig 2B and C). We then exposed OPA1 KO and AA-treated cells to the antioxidant MitoQ, which significantly reduced the microtubule alterations in these cells (Fig 5H and K). As we previously observed (Fig 2), MitoQ treatment had no impact on total dextran uptake (Fig 6A). However, it rescued dextran transport to Rab5-positive EEs and LAMP1-positive lysosomes (Fig 6D–G), consistent with oxidative stress playing a key role in the EE alterations present in cells with mitochondrial dysfunction. Altogether, our results demonstrate that ROS alter EE distribution and subsequent cargo trafficking to lysosomes.

# Discussion

Genetic or chemical alteration in mitochondrial functions leads to muscular and neurological pathologies (3) that were historically associated with ATP deficits. It is now clear that the consequences of mitochondrial defects extend beyond ATP maintenance and include enhanced inflammation and the generation of ROS (4, 5). At the cellular level, mitochondria interact with various organelles to maintain cellular homeostasis. Although functional interactions with the endoplasmic reticulum and lysosomes are well established (28), the influence of mitochondria on other cellular components is less understood. Here, we addressed this question using genetic and chemical models of mitochondrial dysfunction. Our findings revealed that types of mitochondrial dysfunction associated with increased ROS production induce ROS-dependent alterations in centrosomes and microtubules that affect EE distribution and their ability to efficiently deliver cargo to lysosomes.

In physiological conditions, mitochondrial ROS play essential roles in cellular signalling (22). Nevertheless, an imbalance in ROS production and removal can lead to oxidative stress (23, 29). Alterations in mitochondrial structure and function caused by mutations or inhibition of the electron transport chain, as well as the accumulation of damaged mitochondria after the inhibition of mitophagy, increase the production of mitochondrial ROS (30). These ROS can cause cellular damage and contribute to the development of neurodegenerative, metabolic, cardiovascular, and inflammatory diseases (24, 31, 32, 33, 34). Nevertheless, the relationship between ROS signalling and ROS-induced damage is complex and depends on the nature of ROS and its spatiotemporal distribution (23). For example, although nuclear $H_2O_2$ leads to nuclear DNA damage, excess mitochondrial $H_2O_2$ does not (35). Similarly, exposure of cells to high concentrations of $H_2O_2$ causes microtubule depolymerization and loss of architectural stability (36, 37), whereas the lower levels of endogenous ROS produced in our models of mitochondrial dysfunction or GO treatment (8) did not cause microtubule depolymerization.

On the other hand, the elevated ROS present in our models of mitochondrial dysfunction was associated with the presence of supernumerary centrosomes, consistent with a previous report indicating that ROS promote centrosome amplification (24). These centrosomal clusters increase microtubule nucleation and alter intracellular trafficking under physiological conditions in osteoclasts (19). We similarly found that the ROS-dependent centrosome amplification present in cells with mitochondrial dysfunction altered EE trafficking, causing their accumulation around these centrosomes. Importantly, Rab5 aggregation was recapitulated by PLK4-dependent aberrant centrosome duplication and was blocked by inhibiting ROCK1, a kinase that has previously been shown to induce aberrant centrosome duplication in the presence of elevated ROS (27). Altogether, these results suggest that mitochondrial ROS trigger ROCK1-driven centrosome amplification, leading to microtubule alteration and EE clustering. It nevertheless remains possible that other ROS-driven cellular alterations also contribute to this phenotype, including oxidative modifications of centrosomal proteins or microtubules.

Interestingly, the distribution of late endosomes/lysosomes was much less affected by the centrosomal amplification found in our models of mitochondrial dysfunction. This is likely a consequence of the more complex trafficking of these organelles, both anterograde and retrograde, in contrast to the dominant retrograde transport of EEs from the plasma membrane. Consistent with this, although only Rab5-positive vesicles showed perinuclear

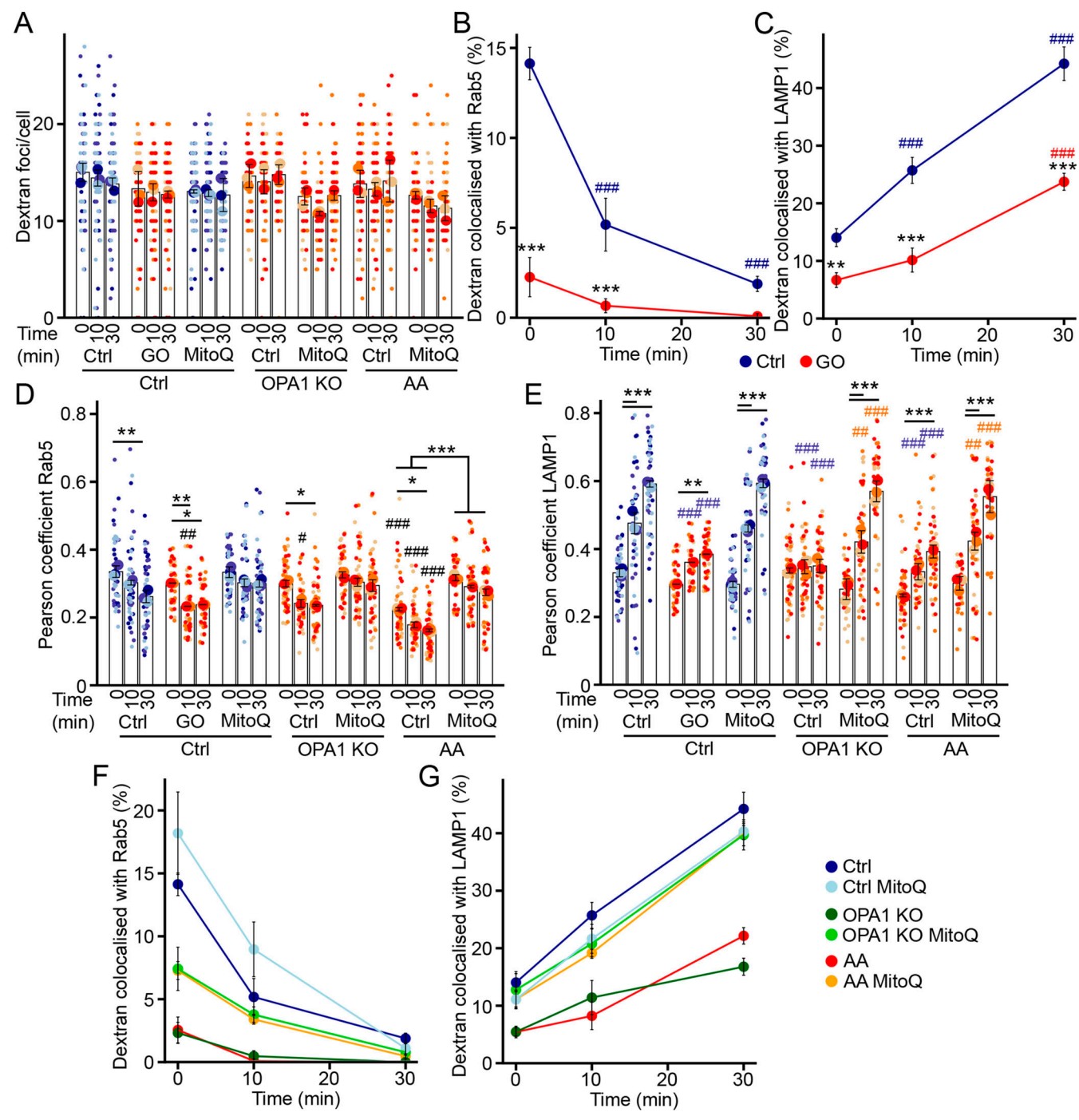

**Figure 6. Antioxidants rescue EE function in cells with mitochondrial dysfunction.**
**(A, B, C)** Dextran trafficking in WT MEFs treated with glucose oxidase (GO). Cells were pulsed with dextran for 5 min, then chased for the indicated times, and total dextran (A) and its colocalization with Rab5 (B) and LAMP1 (C) were quantified by immunofluorescence. Each point represents an independent experiment, with small points in (A) representing individual cells (cells: 82 [Ctrl], 92 [GO], 74 [MitoQ], 73 [KO], 79 [KO + MitoQ], 81 [AA], 76 [AA + MitoQ]). Bars show the average ± SD for three experiments per condition. ***P < 0.001 versus WT, ###P < 0.001 versus 0 min, two-way ANOVA. **(D, E, F, G)** Antioxidants rescue dextran trafficking in OPA1 KO MEFs and AA-treated WT MEFs. Cells were pulsed with dextran for 5 min, then chased for the indicated times, and total dextran (A) and its colocalization with Rab5 (F) and LAMP1 (G) were quantified by immunofluorescence. Each point represents an independent experiment, with small points in (D, E) representing individual cells (51 cells/condition). Bars show the average ± SD for three experiments per condition. ***P < 0.001 versus WT, ###P < 0.001 versus 0 min, two-way ANOVA. All stats are calculated from experiment averages, not cell averages.
Source data are available for this figure.

aggregation and changes in radial speed, LAMP1-positive lyso-somes were less present in the centre of the cytoplasm and had increased angular speed. Overall, these changes in lysosomal behaviour could contribute to the previously reported alterations present in lysosomal structure and function caused by mito-chondrial dysfunction (7, 8).

Pathological conditions because of mitochondrial defects were originally explained by defects in ATP production. However, more recent work has highlighted the multifaceted aspects of mitochondria-related diseases, including several metabolic al-terations not directly related to ATP production and ROS pro-duction, and alterations in organelle contact sites (38, 39, 40). This complex interplay between metabolism and organelle distribution and activity likely plays a major role in the aetiology of mitochondria-related diseases. For example, neurodegenerative diseases are associated with defects in both mitochondria and the endolysosomal compartment, with oxidative stress playing an important role (38, 41). In this context, it is noteworthy that we identified microtubule transport as an important target of ROS that impact EE function and could explain some of the features of these diseases. Nevertheless, as mitochondrial diseases encompass a large array of mutations that affect mitochondrial activities and ROS production in distinct manners, the specific contribution of this ROS-dependent mechanism likely varies across mutations/diseases.

Overall, we propose that the elevated ROS generated by damaged mitochondria cause aberrant centrosome duplication, leading to microtubule alterations that alter EE trafficking and impair their ability to transfer their cargo to lysosomes. Our model is supported by the fact that EE and microtubule defects are re-capitulated in healthy cells by promoting aberrant centrosome duplication or by adding a controlled source of ROS (GO), while being rescued by antioxidants. Our study highlights that mito-chondrial dysfunction not only impacts lysosomes but also in-fluences the function and distribution of EEs, likely contributing to the neuronal impairment caused by mitochondrial dysfunction in neurodegenerative diseases.

# Materials and Methods

Cell culture reagents were bought from Wisent. Other chem-icals were purchased from Sigma-Aldrich, except where indicated.

### Cell culture

WT and OPA1 KO MEFs (gift from Dr. Luca Scorrano, University of Padua), control and DRP1 mutant primary human fibroblasts (42), and HeLa were cultured in DMEM supplemented with 10% FBS. Cells were maintained in an incubator with 5% $CO_2$ until they reached 70–80% confluency before commencing the experiments. Cells were treated as follows: mitochondrial inhibitors antimycin A (50 $\mu$M) or rotenone (5 $\mu$M) for 4 h; antioxidants NAC (10 mM) or MitoQ (100 $\mu$M, # 10-1363; Focus Biomolecules) for 4 h; induction of oxidative stress with 50 milliunits/ml of glucose oxidase for 1 h;

microtubule depolymerization with nocodazole (5 $\mu$M) for 15 min; inhibition of dynein with dynarrestin (2 $\mu$M) for 30 min; $H_2O_2$ treatment (250 $\mu$M) for 30 min; and ROCK1 inhibition with Y-27632 (10 $\mu$M; MedChemExpress) for 3 h.

### Dextran and transferrin uptake experiments

MEFs were grown on coverslips until they reached 70% confluency. Cells were then exposed to 0.5 mg/ml of Dextran, Tetrame-thylrhodamine, 10,000 MW, Lysine Fixable (Fluoro-Ruby) (#D1868; Thermo Fisher Scientific) for 5 min. The media containing dextran were then removed, and fresh media were added. The cells were then chased for 0, 10, and 30 min before being fixed with 4% PFA. For the Tf experiments, the cells were kept in serum-free media on ice for 30 min, then incubated with 0.25 mg/ml of transferrin from human serum (tetramethylrhodamine conjugate, #T2872; Thermo Fisher Scientific) for 5 min. Afterwards, the media containing transferrin were replaced with fresh media, and the cells were chased for 0, 10, and 30 min before being fixed for 10 min with 4% PFA at 25°C. To investigate the endosomal/lysosomal trafficking function under oxidative stress conditions, the cells were first treated with glucose oxidase or MitoQ as above. The media were then changed, and cells were exposed to dextran as above in the presence of GO or MitoQ.

### ROS detection

Protein oxidation was quantified using oxyblots (#S7150; Sigma-Aldrich) according to the manufacturer's instructions. Briefly, cells were lysed in 10 mM Tris–HCl, pH 7.4, 1 mM EDTA, 150 mM NaCl, 1% Triton X-100, complemented with a protease inhibitor cocktail (Sigma-Aldrich) and phosphatase inhibitor (Sigma-Aldrich), kept on ice for 10 min, and centrifuged at 16,000$g$ for 10 min. Protein supernatants were collected, and protein concentration was es-timated by the DC protein assay kit (Bio-Rad). Oxyblot samples (15 $\mu$g) were prepared by adding 5 $\mu$l of 12% SDS (6% final) and treating them with 2,4-dinitrophenylhydrazine (DNPH; +DNPH) or the negative control solution (−DNPH), for 15 min at RT according to the manual. A neutralization solution was then added followed by 1X Laemmli buffer supplemented with ß-mercaptoethanol. Samples were then run on a SDS–PAGE, transferred to nitro-cellulose membranes, and blotted with an antibody against the DNP (2,4-dinitrophenylhydrazone) moiety of the proteins. Membranes were then incubated with a 1:5,000 dilution of horseradish peroxidase–conjugated goat anti-rabbit secondary antibody (Jackson ImmunoResearch) and visualized by en-hanced chemiluminescence (Thermo Fisher Scientific) using a Bio-Rad imaging system.

For MitoSOX, cells were incubated with 5 $\mu$M MitoSOX Red (Life Technologies) for 20 min at 37°C, after which fluorescence was measured at a wavelength of 610 nm in FGP-positive cells using a CytoFLEX (Beckman Coulter).

### Immunofluorescence and cell imaging

Cells were grown on glass coverslips for 24 h before the ex-periments, then treated as indicated, and fixed for 10 min with

4% PFA at 25°C. Cells were permeabilized with 0.2% Triton X-100 in PBS and blocked with 1% BSA/0.1% Triton X-100 in PBS. Cells were then incubated with primary antibodies followed by fluorescently tagged secondary antibodies (1:500; Jackson ImmunoResearch) and DAPI (D1306, 1:100; Invitrogen, Thermo Fisher Scientific). The following primary antibodies were used: mouse anti-$\beta$-tubulin (#T5293; Sigma-Aldrich), rat anti-$\beta$-tubulin (#6161; Abcam), mouse anti-gamma tubulin (#T5326; Sigma-Aldrich), rat anti-LAMP1 (#19992; SCBT), rabbit anti-Rab5 (#3547S, 1:200; Cell Signaling Technologies), rabbit anti-EEA1 (#2411, 1:200; Cell Signaling Technologies), rabbit anti-Rab11 (#3539S, 1:200; Cell Signaling Technologies), and mouse anti-catalase (#66765-1-Ig, 1:200; Proteintech). Imaging was performed using a Leica TCS SP8 confocal microscope fitted with a 63×/1.40 oil objective.

Transfections were done in WT and OPA1 KO MEFs using Metafectene Pro (Biontex). The following plasmids were used: mRFP-Rab5 (plasmid # 14437; Addgene), LAMP1-GFP (plasmid # 16290; Addgene), GFP (pcDNA3-EGFP; plasmid #13031; Addgene), GFP-PLK4 (pEGFP-C3-PLK4-3xFLAG; plasmid #69837; Addgene), and catalase targeted to the mitochondrial matrix by replacing the C-terminal peroxisomal targeting sequence by the N-terminal targeting sequence of ornithine transcarbamylase (gift from Peter S. Rabinovitch, U. Washington (26)). Cells were fixed 24 h later for immunofluorescence. For live-cell imaging, cells were plated after 24 h on glass-bottom dishes in complete medium and grown for another 24 h. The plates were then mounted onto a Leica TCS SP8 confocal microscope fitted with a 63×/1.40 oil objective. Time-lapse images were acquired at a speed of 0.05–0.125 frames/s for 10 min.

### Image processing and analysis

All image manipulation and analysis were done in ImageJ/Fiji. The images shown are from single focal planes unless stated otherwise. Image analysis was done as follows: 1. Images were segmented in ImageJ using Filter/Median (1.0 [in pixels], to reduce noise), then thresholded and adjusted using Binary/Erode. 2. The position of each foci was then identified using the Analyse Particle function with size of 10 px-infinity. Rab5 foci were considered an aggregate when greater than 0.75 $\mu m^2$ (three SD above the average size of WT cells). For the analysis of Rab5 aggregates, cells were considered positive for Rab5 aggregates if they contained at least two particles larger than 0.75 $\mu m^2$ that were within the perinuclear area (defined as ≤30% of the distance between the nuclear membrane and the plasma membrane). For density distributions, the position of all particles (10 px-infinity) was compiled, along with the position of the nucleus, and the plasma membrane was also identified. The data were then fed to a custom R script to calculate the relative distance to the nucleus and the distribution calculated using the Density function. Radial and angular speeds were similarly calculated using a custom R script. Directionality was calculated using the Fiji plugin *Directionality* with Fourier component analysis.

### Data analysis and statistics

Data analysis and statistical procedures were conducted using R. Quantification of immunofluorescence data was performed, and representative images from a minimum of three independent experiments were presented (specific sample sizes are indicated in the respective quantification figures). For experiments where individual cells were quantified, the smaller points represent individual cells in each independent experiment, whereas the larger points represent the average of each experiment. Data are expressed as the mean ± SD per experiment (not cells) as indicated in the figure legends. To assess statistical significance, $t$ test was used for comparisons between two groups, whereas one-way ANOVA with a Tukey post hoc test was employed for multiple comparisons.

## Data Availability

All data that support the conclusions of this work are available within the article and supplemental files.

## Supplementary Information

## Acknowledgements

This work was supported by grants from the Natural Sciences and Engineering Research Council of Canada and the Fondation de l'UQTR. K Todkar was a recipient of a Queen Elizabeth II Diamond Jubilee scholarship and a Fonds du Québec-Santé scholarship. L Chihki was a recipient of Fonds du Québec-Santé scholarships.

### Author Contributions

A Vishwakarma: conceptualization, formal analysis, investigation, methodology, and writing—original draft, review, and editing.
L Chihki: conceptualization, investigation, methodology, and writing—review and editing.
K Todkar: conceptualization, investigation, methodology, and writing—review and editing.
M Ouellet: formal analysis, investigation, methodology, and writing—review and editing.
M Germain: conceptualization, formal analysis, supervision, funding acquisition, project administration, and writing—original draft, review, and editing.

### Conflict of Interest Statement

The authors declare that they have no conflict of interest.

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
