## [Reviewer comments · Life Science Alliance]

Mitochondrial Dysfunction Alters Early Endosomes Trafficking via Microtubule Reorganization

Anjali Vishwakarma, Lilia Chikhi, Kiran Todkar, Mathieu Ouellet, and Marc Germain

DOI: <https://doi.org/10.26508/lsa.202403020>

Corresponding author(s): Marc Germain, Université du Québec à Trois-Rivières

Review Timeline:

Submission Date:	2024-08-28
Editorial Decision:	2024-10-16
Appeal Requested:	2025-04-08
Appeal Decision:	2025-04-08
Revision Received:	2025-08-25
Editorial Decision:	2025-09-22
Revision Received:	2025-09-30
Accepted:	2025-10-02

Scientific Editor: Tim Fessenden

Transaction Report:

October 16, 2024

Re: Life Science Alliance manuscript #LSA-2024-03020-T

Dr. Marc Germain
Université du Québec à Trois-Rivières
Département de Biologie Médicale
3539 LP
3351, boul. des Forges, C.P. 500
Trois-Rivières, Quebec G9A 5H7
Canada

Dear Dr. Germain,

Thank you for submitting your manuscript entitled "Mitochondrial Dysfunction alters Early Endosomes via ROS-Mediated Microtubule Reorganization". The manuscript has been evaluated by expert reviewers, whose reports are appended below. Unfortunately, after an assessment of the reviewer feedback, our editorial decision is against publication in Life Science Alliance.

Although your manuscript is intriguing, I feel that the points raised by the reviewers are more substantial than can be addressed in a typical revision period. If you wish to expedite publication of the current data, it may be best to pursue publication at another journal.

Given the interest in the topic, I would be open to re-submission to Life Science Alliance of a significantly revised and extended manuscript that fully addresses the reviewers' concerns and is subject to further peer review. If you would like to resubmit this work to Life Science Alliance, you may submit an appeal directly through our manuscript submission system. Please note that priority and novelty would be reassessed at re-submission.

Regardless of how you choose to proceed, we hope that the comments below will prove constructive as your work progresses.

Thank you for thinking of Life Science Alliance as an appropriate place to publish your work.

Sincerely,

Reviewer #1 (Comments to the Authors (Required)):

The manuscript by Vishwakarma and colleagues proposes to address the effect of mitochondrial malfunction on the early endosomes.

While the question is very interesting, the data presented falls short of supporting the model proposed.

The main pitfall is that most of the data is correlative. The one experiment that would provide a mechanistic insight suffers from unspecificity in the approach. The term "ROS" is just not acceptable anymore: 20 years ago ROS could be blamed for everything, no mechanism required. Nowadays, the expectations are higher: which ROS, where and how does it impact whatever one is trying to measure. Notably, ROS have also been proposed to diminish lysosomal function (by the lab of Dr. Germain) and have also been proposed to increase lysosomal function (specifically, MCOLN1 activity). If the authors really believe that it is mitochondria-derived ROS that are affecting the cytoskeleton, then a simple experiment is in order: put a H₂O₂ scavenger inside mitochondria (e.g., mito-targeted catalase), or silence SOD1/2 (this will remove mitochondrial H₂O₂ production). This will solve the question if mitochondria-derived H₂O₂ is indeed leading to a cytoskeleton perturbation that eventually affects the transport of early endosomes. However, given that the cytoskeleton is charged with the transport of all organelles in the cell, why would this affect specifically the early endosomes? And also, what does H₂O₂ do to the microtubules? Are they oxidized by Fenton reaction? Are they glutathionylated? This is an important step to understand - even if addressed in an ulterior study, at least the authors ought to provide a discussion regarding how this may happen.

More specific comments:

- in Figure 1E, the authors show a quantification of lysosomal positioning which does not match figure 1A (LAMP1 is clearly

clustered in the peri-nuclear region of OPA1-KO cells; is the quantification wrong or the representative image is not representative?)

- figure 2A has a range of dextran foci that spans from zero to twenty, suggesting that either the analysis method was not sufficiently sensitive to detect the dextrans or that there is an abnormally high heterogeneity in the uptake
- the authors make a big deal of the increase in cells with more than 2 centrosomes upon glucose oxidase treatment of control cells, but the number of cells with two centrosomes under these conditions is only 8% - this clearly would not justify an increase in "blinded" microtubules in 30% of cells or Rab5 perinuclear clustering in 60% of cells - at least in this experiment, these processes do not seem to be related.
- one detail, in the first section of the results, the authors mention that transferrin is recycled back to the plasma membrane. What is recycled is transferrin receptor, not transferrin, which is directed further to the late endosomes/lysosomes.

I cannot give feedback on the tools used to manipulate microtubules or on the interpretation of the microtubule data because these are not within my area of expertise.

Reviewer #2 (Comments to the Authors (Required)):

The manuscript from Vishwakarma and colleagues describes early endosomal (EE)-related alterations in the presence of mitochondrial defects, i.e. upon the KO of the fusion-related protein OPA1 or in the presence of Complex III inhibitor Antimycin A (AA). They show that in the absence of OPA1 or in the presence of AA, EE cannot recycle properly. This seems to be connected with the perinuclear clustering of EE under these conditions, which is due to microtubule bundling. These events seem to be the result of increased oxidative stress, as antioxidants revert these phenotypes.

The manuscript is very interesting and clear. It also provides the first evidence linking mitochondrial dysfunctions and EE. However, there are several major issues that prevent me from recommending this manuscript for publication in the present form.

1. LAMP1-positive vacuoles are hard to see. The distance with Rab5 should be quantified, as Rab5 looks in proximity of the LAMP1 staining.
2. There is a clustering/differential distribution of LAMP1 in KOs too, as they are less dispersed than in WT. Could a cluster be better defined, and is there a parameter that should help to define it or measure it?
3. How can you say that EE recruitment mechanisms are the same between ETC alterations and OPA KO? Are clusters identical? Is protein composition and distribution the same across these two types of clusters? The phenotype could be morphologically identical, yet there is no functional evidence that the molecular mechanisms driving to this phenotype are the same. It would be hard to believe that all sorts of mitochondrial alterations narrow down to EE alterations that can be rescued with MitoQ. I feel that this concept is proposed in an implicit manner throughout the manuscript, and I would suggest to tone it down, add additional mitochondrial alteration paradigms or discuss it thoroughly.
4. How many cells were analyzed for Dextran and Tf trafficking? Why was Pearson's coefficient used? Could the authors comment on the fact that OPA1 KO and AA treatment do not give the same results on Tf colocalization with Rab5? This could also be indicative that the molecular mechanisms beyond these two conditions are not exactly the same (see my previous comment). Plus, Tf colocalization in OPA KO/AA is even higher than in WTs. This is also reported as statistically significant in the graph, yet the authors conclude that the profiles are similar. Instead, similar differences in % are considered highly significant when looking at dextran colocalization. Is there a way to determine the maximal degree of colocalization between Dextran or Tf and Rab5 in MEFs, so that appropriate comparisons can be made between WT, OPA1KO and AA?
5. When analyzing foci, how was the 0.75 μm minimal size chosen? Plus, a median filter of 1.0 induces a strong effect on the image rendering. Why such a strong filter is needed? What would happen if this filter is removed? What is considered "perinuclear" and what is not perinuclear? I wonder if there is a minimal or maximal distance determining which is which. Since the R script and these features are not disclosed in the materials and methods, this is hard to determine.
6. Why the gammaTubulin staining appears to be located in the cytosol in WT MEFs? This is expected to be a centrosomal-only marker. Is this staining due to the bleedthrough of the beta-Tubulin antibody?
7. It is hard to quantitatively appreciate microtubules being bundled. I believe that a more refined characterization of cytoskeletal complexity (e.g. skeletonization, distribution, radiality etc.) should be proposed in addition to cell number/%.
8. What happens with the addition of GO to OPA1 KO/AA in terms of centrosomal number? In other words, do we know whether OPA1 KO/AA are conditions with the maximum of tolerance concerning centrosomal outnumber, or a maximum of 9% does not have any functional consequences to parameters such as cell division, for instance?
9. A formal proof of increased oxidative stress levels in OPA1 KO/AA should be provided
10. Is microtubule bundling promoting increased oxidative stress levels in OPA1 KO/AA? The authors propose that oxidative stress cause microtubule bundling, but do not exclude that microtubule bundling causes oxidative stress.

Reviewer #3 (Comments to the Authors (Required)):

The authors describe the effects of mitochondrial dysfunction on early endosome (EE) trafficking by using genetic and pharmacological approaches to perturb mitochondrial function and note various observations that relate to trafficking. These

observations include EE clustering in the perinuclear region, reduced rates of trafficking of the fluid-phase marker dextran through EE to reach lysosomes, as well as partial effects on Tf recycling. In addition, the authors show some cytoskeletal alterations such as centrosome duplication and dense microtubule staining. The authors were able to recapitulate these findings in WT cells treated with glucose oxidase that generates ROS and were able to revert EE positioning and cytoskeletal abnormalities with antioxidants. The authors conclude that ROS generated as a result of mitochondrial dysfunction causes cytoskeletal abnormalities that in turn drive clustering of EEs, affecting cargo trafficking through this compartment. While this study does indeed show disruptions to EE trafficking as a result of mitochondrial dysfunction and that these effects are attributable to oxidative stress, the overall findings are unfortunately neither interesting nor novel, for the following reasons:

1) Oxidative stress that results from dysfunctional mitochondria would incur widespread changes/damage to cells by oxidizing proteins, lipids, DNA and any other molecules susceptible to oxidation, affecting signaling pathways, genomic stability etc. In fact, peroxidation of membrane phospholipids would likely have major effects on membrane trafficking by altering membrane fluidity, the activities of channels and so on. Yet, this very relevant consideration is not mentioned in the manuscript. Reviews such as PMID: 38370049 highlight these points well: that almost everything in the cell stands to be affected by ROS.

2) Given the widespread effects that oxidative stress would incur on cells, it is difficult to understand the rationale for looking at EE specifically. In addition, the effects of ROS on EE trafficking has been studied by at least one other group, Kano et al (PMID 21277337), showing endocytosis defects upon treatment of cells with H₂O₂. This paper should have been discussed, especially since the Kano et al study does not show EE clustering upon H₂O₂ treatment - could this be due to the different method/amount/duration of oxidative stress? Either way, the current manuscript is not the first to report defects in EE trafficking as a result of oxidative stress.

3) Finally, both the microtubule densification and the presence of supernumerary centrosomes due to elevated ROS are phenotypes that have already been described by Goldblum et al, 2021 and Chae et al, 2005, respectively. Thus, it seems that that most of the findings of the current manuscript are unfortunately not novel.

To conclude, this lack of specificity of oxidative stress (i.e. ROS would affect pretty well all cellular pathways) combined with lack of novelty make the manuscript rather underwhelming and I feel it does not contribute to the field in a meaningful way.

Dear Dr. Sawey,

In your decision letter following the initial round of review, you requested us to fully address the reviewers' concerns and significantly revise the manuscript in order for you to consider a resubmission of our manuscript. We have now performed all of the requested experiments and significantly revised the manuscript.

Importantly, we have strengthened the mechanistic relationship between reactive oxygen species (ROS) and the early endosome alterations we find in our models of mitochondrial dysfunction. We also clarified the issues concerning the methodology and interpretation of the data that were raised by the reviewers. Altogether, this significantly revised and extended version of our manuscript demonstrates that mitochondrial dysfunction leads to the disruption of early endosome trafficking as a consequence of ROS-dependent alterations in centrosomes and microtubules. As it is now clear that the cellular alterations leading to the clinical presentation of mitochondrial diseases go well beyond ATP production, our work provides important insights into the underlying molecular events.

Specifically, we now provide more mechanistic evidence linking ROS to the endosomal phenotype we describe in cells with mitochondrial dysfunction and show that it is specifically H₂O₂-dependent (Reviewer 1). We also addressed all of Reviewer 2's questions by providing new data and rewriting the text to clarify the methodology. Then, to better demonstrate that early endosome alterations are the consequence of a specific ROS signalling pathway, not general ROS-induced damage (reviewer 3), we provide new data showing that it can be recapitulated by altering centrosomes without ROS production, that it can be blocked with a ROCK1 inhibitor, and that a dose of H₂O₂ causing a large increase in oxidized proteins does not lead to early endosome alterations. Finally, we reworked the text to better highlight the novelty of our work (Reviewer 3), which is to demonstrate that ROS-generating mitochondrial dysfunction leads to impaired endosomal trafficking (as highlighted by Reviewer 2). On this last point, while we have improved the description of the novelty of our work, we do think that Reviewer 3's request for novelty goes well beyond what is required by the aims and scope of LSA, a point for which we hope you will agree with us. You can find a point-by-point response to the reviewers' comments in the attached response to the reviewers. We also attached a cover letter and the revised manuscript.

April 11, 2025

MS: LSA-2024-03020-T

Dr. Marc Germain
Université du Québec à Trois-Rivières
Département de Biologie Médicale
3539 LP
3351, boul. des Forges, C.P. 500
Trois-Rivières, Quebec G9A 5H7
Canada

Dear Dr. Germain,

I recently took over for Eric Sawey as Executive Editor of LSA. I have gone over this appeal request, the rebuttal letter, and the original manuscript. I am pleased to let you know that we have decided to send your manuscript for external review. Please forgive the time needed to assess this appeal while I am getting up to speed here at LSA.

Please use the following link to submit your manuscript:

<https://lsa.msubmit.net/cgi-bin/main.plex?el=A4Na5BUL3A6CGLS211B9ftdLICIQj8WarZWiohVos1kwZ>

Yours sincerely,

Tim Fessenden
Life Science Alliance

Response to the reviewers (Reviewers' comments are in **bold**)

Reviewer #1 (Comments to the Authors (Required)):

The manuscript by Vishwakarma and colleagues proposes to address the effect of mitochondrial malfunction on the early endosomes.

While the question is very interesting, the data presented falls short of supporting the model proposed.

The main pitfall is that most of the data is correlative. The one experiment that would provide a mechanistic insight suffers from unspecificity in the approach. The term "ROS" is just not acceptable anymore: 20 years ago ROS could be blamed for everything, no mechanism required. Nowadays, the expectations are higher: which ROS, where and how does it impact whatever one is trying to measure. Notably, ROS have also been proposed to diminish lysosomal function (by the lab of Dr. Germain) and have also been proposed to increase lysosomal function (specifically, MCOLN1 activity). If the authors really believe that it is mitochondria-derived ROS that are affecting the cytoskeleton, then a simple experiment is in order: put a H₂O₂ scavenger inside mitochondria (e.g., mito-targeted catalase), or silence SOD1/2 (this will remove mitochondrial H₂O₂ production). This will solve the question if mitochondria-derived H₂O₂ is indeed leading to a cytoskeleton perturbation that eventually affects the transport of early endosomes. However, given that the cytoskeleton is charged with the transport of all organelles in the cell, why would this affect specifically the early endosomes? And also, what does H₂O₂ do to the microtubules? Are they oxidized by Fenton reaction? Are they glutathionylated? This is an important step to understand - even if addressed in an ulterior study, at least the authors ought to provide a discussion regarding how this may happen.

The reviewer brings up a number of important questions which we have now addressed:

a) **The main pitfall is that most of the data is correlative.**

We have now conducted several new experiments to better delineate the mechanism through which mitochondrial ROS lead to microtubule alterations and EE aggregation. First, we overexpressed PLK4 to induce centrosome duplication and microtubule bundling and showed that this leads to EE aggregation (New Figure 4G, New sup. Figure 3) in the absence of increased ROS (New Sup. Figure 5E), consistent with the idea that microtubule bundling causes EE trafficking defects. Second, as ROCK1 was previously shown to promote centrosome duplication in a ROS-dependent manner (PMID: 29768267), we determined the effect of ROCK1 inhibition on microtubule bundling and EE aggregation. Consistent with our model, ROCK1 inhibition rescued both phenotypes (New sup. Figure 5F-G). We also performed the mito-catalase experiment suggested by the reviewer. In our hand, the construct we used (from PMID: 15879174) targeted catalase to mitochondria in 50% of the cells, the other 50% showing strong cytosolic staining (New Sup. Figure 5B). This allowed us to test both the role of matrix-generated H₂O₂ and the effect of quenching overall H₂O₂ levels. Cytosolic catalase completely rescued both microtubule alterations and EE aggregation (New Sup. Figure 5C-D), confirming that the alterations are H₂O₂-dependent. Matrix-targeted catalase also rescued microtubules and EE aggregation, albeit at a lower extent (New Sup. Figure 5C-D). We think that this is because while H₂O₂ is

produced on both sides of the inner membrane (matrix and intermembrane space), matrix-catalase only quenches matrix H_2O_2 , leaving a significant amount of H_2O_2 which can then diffuse to the cytosol to affect microtubules. This would also be consistent with the fact that the mitochondria-targeted antioxidant MitoQ rescues all aspects of the endosomal phenotype present in OPA1 KO cells (Figures 5-6). Overall, our results support the idea that mitochondrial ROS (H_2O_2) stimulates ROCK1-dependent centrosome amplification that leads to EE aggregation and endosome trafficking defects.

b) Given that the cytoskeleton is charged with the transport of all organelles in the cell, why would this affect specifically the early endosomes?

We agree with the reviewer that a widespread reorganisation of microtubule should lead to changes in organelles other than EEs. This issue was not clearly stated in the previous version of the manuscript, which we have now corrected. Specifically, lysosomes are less present in the middle of the cytoplasm (away from the nucleus and the plasma membrane) in OPA1 KO cells (Sup. Figure 1A). This is accompanied by a change in their angular speed (Figure 3C). This is now better acknowledged in the text. These alterations are nevertheless less dramatic than for EEs. We think this is because EEs are mainly dependent on retrograde transport while other organelles such as lysosomes use both anterograde and retrograde transport, which are both affected by microtubule bundling.

c) What does H_2O_2 do to the microtubules?

As mentioned above, our results suggest that ROS acts through ROCK1 to stimulate centrosome duplication (New Sup. Figure 5F-G). However, it remains possible that ROS affect directly (oxidation, glutathionylation) or indirectly (through other signalling pathways) other components of this pathway including centrosomes, microtubules or possibly endosomal proteins. This is now discussed in the text.

More specific comments:

1. in Figure 1E, the authors show a quantification of lysosomal positioning which does not match figure 1A (LAMP1 is clearly clustered in the peri-nuclear region of OPA1-KO cells; is the quantification wrong or the representative image is not representative?)

The reviewer is right that there are some alterations in lysosome positioning in OPA1 KO cells (as seen by the density distribution in Supplemental Figure 1A) and the previous version of the manuscript was not clear about this. There are nevertheless a few points to clarify here:

First, the image we used was not really representative of the overall distribution of LAMP1 vesicles. We have thus changed it (New Figure 1A).

Second, what we meant by perinuclear clustering was not clearly defined in the text, leading to some confusion. What we actually measured as clustering was the aggregation of the vesicles in the perinuclear region, not simply an increase in their presence closer to the nucleus. These aggregates were defined as perinuclear vesicles that had a size 3 SD greater than the average for the control cells. The method has been updated to clarify this and we have changed “clusters” for “aggregates” throughout the text and figures. In that sense, LAMP1-positive vesicles do not form perinuclear aggregates (although their overall distribution is altered with more at the periphery of the cell, which has now been clarified in the text).

2. figure 2A has a range of dextran foci that spans from zero to twenty, suggesting that either the analysis method was not sufficiently sensitive to detect the dextrans or that there is an abnormally high heterogeneity in the uptake

While we cannot exclude that some of the dextran failed to be detected in our assay conditions, our results are similar in cell-to-cell variability to previously published data (protocol in PMID: 35530513, research paper in PMID: 34165494). It is thus unclear to us what would constitute an appropriate level of variability for the reviewer, especially since actual cell-to-cell variability is rarely shown in these experiments (usually only the experiment average is shown, which corresponds to the larger dots in Figure 2A from which we generated the stats).

Importantly, while there is some (normal, we would argue) variability in the amount of dextran each cell takes up, the results are similar across experiments, highlighting the robustness of our analysis. It is also important to note that we base our statistical analysis and conclusions on the average per experiment (this has been clarified in the text), which shows limited variability, consistent with what others have shown.

3. the authors make a big deal of the increase in cells with more than 2 centrosomes upon glucose oxidase treatment of control cells, but the number of cells with two centrosomes under these conditions is only 8% - this clearly would not justify an increase in "blunted" microtubules in 30% of cells or Rab5 perinuclear clustering in 60% of cells - at least in this experiment, these processes do not seem to be related.

We think that this apparent discrepancy is in the way the quantification was shown in the figure rather than a lack of causal effect.

In an attempt clearly show the effect of oxidative stress, the figure (now 5C) only presented the data for 3 and more centrosomes, which corresponds to a minority of cells, hence the small numbers. To clarify this point and provide a better analysis of centrosome alterations caused by GO, we now provide the full distribution of centrosome numbers (New Figure 5C) showing a change in distribution similar to what was observed in OPA1 KO and AA-treated cells (Figure 4B). As the cells with 2 centrosomes could represent dividing cells, we also measured the distance between centrosomes and found that, similar to OPA1 KO and AA-treated cells, centrosomes were much closer in GO-treated cells than control cells (New Figure 5D). Thus, about 60% of the GO-treated cells have at least two centrosomes and these are close together, promoting microtubule bundling and Rab5 clustering.

In terms of microtubule bundling, as Reviewer 2 pointed out (see Reviewer 2 point 7), our original analysis was not very refined. To address this, we have now included a measure of microtubule directionality based on the dispersion of the microtubule signal (the more bundled the microtubules are, the lower is the dispersion – the SD of the signal). This is exactly what we observe in a majority of OPA1 KO, AA-treated and GO-treated cells (New Figures 4F and 5F). This again supports the idea that microtubules are altered in most GO-treated cells, consistent with the Rab5 aggregation we observed in these cells (Figure 5G).

4. one detail, in the first section of the results, the authors mention that transferrin is recycled back to the plasma membrane. What is recycled is transferrin receptor, not transferrin, which is directed further to the late endosomes/lysosomes.

In fact, it is both the receptor and transferrin that are recycled to the plasma membrane after delivering iron to early endosomes (PMID: 23046645, PMID: 21968002). This is why

labelled transferrin it is widely used to study recycling endosomes (otherwise, it would label lysosomes, not recycling endosomes).

I cannot give feedback on the tools used to manipulate microtubules or on the interpretation of the microtubule data because these are not within my area of expertise.

Reviewer #2 (Comments to the Authors (Required)):

The manuscript from Vishwakarma and colleagues describes early endosomal (EE)-related alterations in the presence of mitochondrial defects, i.e. upon the KO of the fusion-related protein OPA1 or in the presence of Complex III inhibitor Antimycin A (AA). They show that in the absence of OPA1 or in the presence of AA, EE cannot recycle properly. This seems to be connected with the perinuclear clustering of EE under these conditions, which is due to microtubule bundling. These events seem to be the result of increased oxidative stress, as antioxidants revert these phenotypes.

The manuscript is very interesting and clear. It also provides the first evidence linking mitochondrial dysfunctions and EE. However, there are several major issues that prevent me from recommending this manuscript for publication in the present form.

1. LAMP1-positive vacuoles are hard to see. The distance with Rab5 should be quantified, as Rab5 looks in proximity of the LAMP1 staining.

We have changed the image to better represent both the presence of LAMP1 vacuoles and the fact that Rab5- but not LAMP1-positive vesicles aggregate close to the nucleus. We also did the analysis requested by the reviewer, but did not find a significant difference in the distance between lysosomes and EEs in WT and OPA1 KO cells, consistent with mostly Rab5-positive vesicles being redistributed towards the nucleus in the KO cells (Figure 1 for reviewers).

Figure 1 for reviewers.
Distance between Rab5 and LAMP1 vesicles

2. There is a clustering/differential distribution of LAMP1 in KOs too, as they are less dispersed than in WT. Could a cluster be better defined, and is there a parameter that should help to define it or measure it?

The distribution of LAMP1-positive lysosomes is indeed somewhat altered in OPA1 KO MEFs (Supplementary Figure 1A), although not to the extent suggested by the original Figure 1A. It was not representative of the actual phenotype and was thus changed in the new version of the manuscript (also see Reviewer 1 Point 1).

In terms of clustering, what we actually measured is the aggregation of the vesicles in the perinuclear region (the presence of larger vesicles that accumulate close to the nucleus – see Reviewer 1 Point 1). While lysosomes position is partially altered in the mutant cells (with more at the periphery of the cell), we do not see this type of aggregation. To clearly

distinguish these points, we now label the Rab5 clusters as aggregates throughout the text and figures.

3. How can you say that EE recruitment mechanisms are the same between ETC alterations and OPA KO? Are clusters identical? Is protein composition and distribution the same across these two types of clusters? The phenotype could be morphologically identical, yet there is no functional evidence that the molecular mechanisms driving to this phenotype are the same. It would be hard to believe that all sorts of mitochondrial alterations narrow down to EE alterations that can be rescued with MitoQ. I feel that this concept is proposed in an implicit manner throughout the manuscript, and I would suggest to tone it down, add additional mitochondrial alteration paradigms or discuss it thoroughly.

Our original perspective was that a large spectrum of mitochondrial alterations leads to increased ROS production. These alterations should thus cause at least some level of microtubule/EE alterations. This is supported by our observation that distinct alterations leading to ROS production cause microtubule bundling and EE aggregation, while ROS scavengers prevent it. However, as the reviewer points out, distinct mitochondrial alterations can have different mechanisms/outcomes that might not rely heavily on ROS production. While we still think that the type of alterations we describe here will be present in cells with chronically increased ROS, some other mitochondrial alterations that do not significantly increase ROS (like mutants in metabolic enzymes) might not show this phenotype. This is now discussed in the text.

4. How many cells were analyzed for Dextran and Tf trafficking? Why was Pearson's coefficient used? Could the authors comment on the fact that OPA1 KO and AA treatment do not give the same results on Tf colocalization with Rab5? This could also be indicative that the molecular mechanisms beyond these two conditions are not exactly the same (see my previous comment). Plus, Tf colocalization in OPA KO/AA is even higher than in WTs. This is also reported as statistically significant in the graph, yet the authors conclude that the profiles are similar. Instead, similar differences in % are considered highly significant when looking at dextran colocalization. Is there a way to determine the maximal degree of colocalization between Dextran or Tf and Rab5 in MEFs, so that appropriate comparisons can be made between WT, OPA1KO and AA?

a) **How many cells were analyzed for Dextran and Tf trafficking?** Around 30 cells/experiment were analysed for the colocalization analysis, while this was 18 cells/experiment for Pearson's coefficient. We added the actual numbers in the figure legends.

b) **Why was Pearson's coefficient used?** We wanted to have two independent manners to measure the colocalization between the endocytosed cargo and endosomes. So, in addition to the manual analysis presented in Figure 2, we used Pearson's coefficient because 1. This has been widely used in the field to measure colocalization (PMID: 24034251, PMID: 38619530) and 2. Contrary to Manders' coefficient, it works with the raw pixel values, making it more robust for this kind of analysis.

c) **Could the authors comment on the fact that OPA1 KO and AA treatment do not give the same results on Tf colocalization with Rab5?** Our interpretation of Figure 2E

was that while the averages for AA and OPA1 KO are somewhat different for the colocalization with Rab5, the variability of the AA data makes it difficult to conclude a distinct behaviour.

We nevertheless did as suggested by the reviewer and normalised all the Dextran and Tf data on the amount of colocalization in WT cells at $t=0$ (the closest we could get to maximum colocalization). The results (Figure 2 for reviewers) show that the two cargo (Tf – solid lines; Dextran dashed lines) behave completely distinctly as expected. In this analysis, the difference between AA and OPA1 KO Tf colocalization at $t=0$ is significant.

While this could be consistent with distinct underlying mechanisms, the difference could also potentially be explained by other

factors (4-hour treatment for AA vs long-term KO for OPA1 for example). In addition, none of our other measures (Rab5 colocalization as analysed using Pearson's coefficient (Sup. Figure 2C); total Tf (Figure 2D)) showed significant differences between OPA1 KO cells and AA-treated cells, making it less likely that this represents an actual mechanistic difference.

We have acknowledged in the text that OPA1 KO and AA-treated cells show higher colocalization between Tf and Rab5 than control cells. We also mentioned the difference in colocalization with Rab11 and the reduced recycling of Tf to conclude that Tf recycling is somewhat affected in OPA1 KO and AA-treated cells. We did not want to comment further on Tf transport as the trafficking issues present in OPA1 KO and AA-treated cells are more subtle and complex than for dextran delivery to lysosomes, which we focussed on for the rest of the paper. This has been clarified this in the text.

5. When analyzing foci, how was the 0.75 μm minimal size chosen? Plus, a median filter of 1.0 induces a strong effect on the image rendering. Why such a strong filter is needed? What would happen if this filter is removed? What is considered "perinuclear" and what is not perinuclear? I wonder if there is a minimal or maximal distance determining which is which. Since the R script and these features are not disclosed in the materials and methods, this is hard to determine.

The median filter of 1.0 means one pixel, not 1 μm . This is the minimal size to which the median can be applied. It does not alter any of the actual structures we measure but will remove a lot of the noise in the images. Sorry if this was not clear. An example image before and after median is applied is shown in Figure 3 for reviewers.

We defined the 0.75 μm^2 size as 3 SD above the average size of Rab5 vesicles in WT cells. We defined the perinuclear region as the area $\leq 30\%$ of the distance between the nucleus

Figure 2 for reviewers. Comparison of dextran and Tf colocalization with Rab5. The data from Figures 2B and 2E was normalised to WT controls ($t=0$). Dextran, Dashed lines; Tf, Solid lines.

and the plasma membrane. 85% of the Rab5 vesicles larger than $0.75 \mu\text{m}^2$ were contained within this region. Cells were considered positive for Rab5 aggregates if they contained at least two of these foci within the perinuclear region. This has been clarified in the methods.

Figure 3 for reviewers.
Example of median processing of Rab5 images.

6. Why the gammaTubulin staining appears to be located in the cytosol in WT MEFs? This is expected to be a centrosomal-only marker. Is this staining due to the bleedthrough of the beta-Tubulin antibody?

There is no bleedthrough of other antibodies in this experiment. However, the antibody does sometimes show some cytosolic or background staining, as has also been observed by others (PMID: 24098540, PMID: 30271923).

7. It is hard to quantitatively appreciate microtubules being bundled. I believe that a more refined characterization of cytoskeletal complexity (e.g. skeletonization, distribution, radially etc.) should be proposed in addition to cell number/%.

As suggested by the reviewer, we have reanalysed our microtubule images using microtubule directionality. This measures the dispersion of the microtubule signal. In this assay, a decrease in the dispersion of the fluorescent signal of microtubules (i.e. bundling) is seen as a decrease in the standard deviation (SD) of the signal. Using this method, we confirmed the alterations we found in our models of mitochondrial dysfunction (New Figure 4F) and after GO exposure (New Figure 5F), as well as the rescue by antioxidants (New Figure 5K).

8. What happens with the addition of GO to OPA1 KO/AA in terms of centrosomal number? In other words, do we know whether OPA1 KO/AA are conditions with the maximum of tolerance concerning centrosomal outnumber, or a maximum of 9% does not have any functional consequences to parameters such as cell division, for instance?

As suggested by the reviewer, we measured the effect of GO on centrosome number in OPA1 KO and AA-treated cells.

Figure 4 for reviewers.
Quantification of centrosome numbers in OPA1 KO and AA-treated cells exposed to GO

Adding GO to these cells did not have any effect on centrosome numbers, suggesting that it does not further increase oxidative stress response in these cells (Figure 4 for reviewers). In terms of cells division, OPA1 KO MEFs are known to grow slowly, which has been presumed to be related to their metabolic alterations.

9. A formal proof of increased oxidative stress levels in OPA1 KO/AA should be provided

We previously published mitoSOX data for the same cells (Demers-Lamarche (2016) J. Biol. Chem) so we did not repeat the experiment here. We have nevertheless addressed the reviewer's comment by providing Oxyblots showing an increase in oxidized proteins in OPA1 KO cells and WT cells treated with AA (New Figure 5A, Quantification in New sup. Figure 4A).

10. Is microtubule bundling promoting increased oxidative stress levels in OPA1 KO/AA? The authors propose that oxidative stress cause microtubule bundling, but do not exclude that microtubule bundling causes oxidative stress.

We addressed this question by overexpressing PLK4, a kinase required for centrosome biogenesis that leads to aberrant centrosome duplication when overexpressed. Consistent with this, PLK4 expression caused microtubule bundling (New sup. Figure 3) and Rab5 aggregation (New Figure 4G). However, it did not cause an increase in ROS as measured using MitoSOX (New sup. Figure 5E), indicating that microtubule bundling does not directly increase ROS.

Reviewer #3 (Comments to the Authors (Required)):

The authors describe the effects of mitochondrial dysfunction on early endosome (EE) trafficking by using genetic and pharmacological approaches to perturb mitochondrial function and note various observations that relate to trafficking. These observations include EE clustering in the perinuclear region, reduced rates of trafficking of the fluid-phase marker dextran through EE to reach lysosomes, as well as partial effects on Tf recycling. In addition, the authors show some cytoskeletal alterations such as centrosome duplication and dense microtubule staining. The authors were able to recapitulate these findings in WT cells treated with glucose oxidase that generates ROS and were able to revert EE positioning and cytoskeletal abnormalities with antioxidants. The authors conclude that ROS generated as a result of mitochondrial dysfunction causes cytoskeletal abnormalities that in turn drive clustering of EEs, affecting cargo trafficking through this compartment.

While this study does indeed show disruptions to EE trafficking as a result of mitochondrial dysfunction and that these effects are attributable to oxidative stress, the overall findings are unfortunately neither interesting nor novel, for the following reasons:

1) Oxidative stress that results from dysfunctional mitochondria would incur widespread changes/damage to cells by oxidizing proteins, lipids, DNA and any other molecules susceptible to oxidation, affecting signaling pathways, genomic stability etc. In fact, peroxidation of membrane phospholipids would likely have major effects on

membrane trafficking by altering membrane fluidity, the activities of channels and so on. Yet, this very relevant consideration is not mentioned in the manuscript. Reviews such as PMID: 38370049 highlight these points well: that almost everything in the cell stands to be affected by ROS.

We agree with the reviewer that ROS can cause widespread damage to cellular structures. However, as Reviewer 1 pointed out, the situation is much more complex than that and the outcome depends on the nature, timing, location and amount of ROS. A large supraphysiological increase in ROS (i.e. exogenous H₂O₂) damages cells and can lead to cell death. However, lower levels of ROS (especially endogenous H₂O₂) act as signalling molecules that are required for normal physiology (see PMID: 24845678 for a discussion of the signalling vs damaging roles of ROS). It was also recently shown that mitochondria-derived ROS does not lead to nuclear DNA damage (PMID: 38548751), further supporting the idea that ROS are much more than molecules that non-selectively damage cellular content.

Our results are totally consistent with this. First, the amount of oxidation present in our cellular models is still relatively low relative to a single dose of exogenous H₂O₂, as seen in New Figure 5A and New sup Figure 4A, where adding H₂O₂ directly to cells causes a much greater increase in oxidized proteins than impairing mitochondrial activity with AA or in OPA1 KO cells. This is an important point as numerous studies (including several cited by this reviewer) have used exogenous H₂O₂ to model oxidative stress which, for the reasons cited above, is not necessarily representative of the endogenous ROS that were present in our study.

Second, we now show that the EE phenotype can be recapitulated by overexpressing the centrosome-associated kinase PLK4, leading to aberrant centrosome duplication, microtubule bundling and EE aggregation in the absence of elevated ROS (New Figure 4G, New sup. Figures 3 and 5E). The EE phenotype of OPA1 KO cells can also be rescued by blocking ROCK1 (New sup. Figure 5F-G), a kinase that promotes centrosome duplication in response to ROS (PMID: 29768267). This further highlights the existence of a selective signalling pathway linking ROS production to microtubule alterations and EE aggregation. Overall, we have now added several new experiments demonstrating that the EE defects we observed are not the consequence of general non-specific ROS-driven oxidation of proteins and lipids, but rather results from the activation of a selective ROS/ROCK1-dependent signalling pathway promoting aberrant centrosome duplication. We also modified the text to better explain these results and highlight the different roles of ROS.

2) Given the widespread effects that oxidative stress would incur on cells, it is difficult to understand the rationale for looking at EE specifically. In addition, the effects of ROS on EE trafficking has been studied by at least one other group, Kano et al (PMID 21277337), showing endocytosis defects upon treatment of cells with H₂O₂. This paper should have been discussed, especially since the Kano et al study does not show EE clustering upon H₂O₂ treatment - could this be due to the different method/amount/duration of oxidative stress? Either way, the current manuscript is not the first to report defects in EE trafficking as a result of oxidative stress.

As suggested by the reviewer, we have repeated the Kano et al. H₂O₂ experiment in our cell model. Consistent with their findings, we do not see centrosome duplication, microtubule bundling or Rab5 aggregation following H₂O₂ treatment (New Sup. Figure 4).

This is consistent with what we discussed in point 1 above, that a large and sudden increase in ROS affects cells differently than a small but sustained increase in ROS. This also supports our view that the effect of ROS on endosomal transport is the result of a selective process, not some random damage caused by ROS.

Concerning the novelty, the specific contribution of our study is to show that mitochondrial dysfunction leads to impaired endosomal trafficking, which had not been reported before (as also stated by Reviewer 2 in its introductory remarks). We used previous reports linking (mostly) exogenous H₂O₂ to EEs or microtubules defects to give us insights that allowed us to define the mechanism through which mitochondrial defects lead to EE alterations. We have also added the references that the reviewer mentioned as missing in the original manuscript.

3) Finally, both the microtubule densification and the presence of supernumerary centrosomes due to elevated ROS are phenotypes that have already been described by Goldblum et al, 2021 and Chae et al, 2005, respectively. Thus, it seems that that most of the findings of the current manuscript are unfortunately not novel.

Again, the main message and novelty of our work is that these processes connected and affect cells with mitochondrial dysfunction, leading to alterations in endosome function. The papers cited by the reviewer only concern the direct relationship between ROS and either centrosomes or microtubules, they do not link these concepts together in the context of mitochondrial dysfunction. Here, we built on this previous knowledge to identify the mechanism through which mitochondrial dysfunction alters endosomes (which this reviewer acknowledged that we did convincingly, which we appreciate). We have reworked the discussion to better address the reviewer's points. We also thank the reviewer for pointing out some papers that we failed to include in the original version of the manuscript and are now properly cited.

Overall, we think that our manuscript clearly fits within the Aims and scope of Life Science Alliance relative to its novelty and we hope that, with the new experiments we now provide and the clarifications provided above and in the manuscript, this reviewer will agree.

To conclude, this lack of specificity of oxidative stress (i.e. ROS would affect pretty well all cellular pathways) combined with lack of novelty make the manuscript rather underwhelming and I feel it does not contribute to the field in a meaningful way.

Concerning ROS: As mentioned above, ROS signalling is much more selective than the reviewer implies (see Reviewer 1 and PMID: 24845678). To support this, we now provide new data indicating that a single dose of exogenous H₂O₂ does not recapitulate our finding, arguing for their selectivity. Furthermore, we provide new data mechanistically linking mitochondrial ROS to microtubule alterations and EE aggregation. Thus, the mechanism we propose here is selective, and not the result of non-specific damage caused by ROS.

Concerning the novelty: The key finding of our work is that mitochondrial dysfunction alters early endosomes (not that ROS affect centrosomes or microtubules), which has major repercussions for our understanding of mitochondrial diseases (see Reviewer 2). Our work builds on previous studies addressing a specifically aspect of the relationship between ROS and centrosome or microtubules to define the mechanism by which mitochondrial dysfunction leads to EE alterations. We thus think that this is clearly within the scope of Life Science Alliance relative to novelty.

September 22, 2025

RE: Life Science Alliance Manuscript #LSA-2024-03020-TR-A

Dr. Marc Germain
Université du Québec à Trois-Rivières
Département de Biologie Médicale
3539 LP
3351, boul. des Forges, C.P. 500
Trois-Rivières, Quebec G9A 5H7
Canada

Dear Dr. Germain,

Thank you for submitting your revised manuscript entitled "Mitochondrial Dysfunction alters Early Endosomes via ROS-Mediated Microtubule Reorganization". As you will see, the reviewers are satisfied with the changes in place. Please consider the remaining suggestions from Reviewer 3. We would be happy to publish your paper in Life Science Alliance pending those changes and final revisions necessary to meet our formatting guidelines.

- Please upload your main manuscript text as an editable doc file.
- Please upload all figure files as individual ones, including the supplementary figure files; all figure legends should only appear in the main manuscript file.
- Please add the X and Bluesky handles of your host institute/organization, as well as your own and/or one of the authors, in our system.
- Please consider changing the title for improved clarity: "Mitochondrial dysfunction impairs early endosome trafficking via microtubule reorganization" or "Oxidative stress impairs early endosome trafficking via microtubule reorganization". Please also ensure the titles in the system and the manuscript file are consistent with each other.
- The "Data Availability" section should be placed after the Materials & Methods section. Please consult our guidelines at <https://www.life-science-alliance.org/manuscript-prep#format>
- Please add an Author Contributions section to your main manuscript text.
- Please add a Conflict of Interest statement to your main manuscript text.
- Please add your main, supplementary figure, and table legends to the main manuscript text after the references section.
- The images in Figure 5B, panel "GO", and 4D, panel "AA" appear identical. Please verify these are correct.

A. FINAL FILES:

- An editable version of the final text (.DOC or .DOCX) is needed for copyediting (no PDFs).
- High-resolution figure, supplementary figure and video files uploaded as individual files: See our detailed guidelines for preparing your production-ready images, <https://www.life-science-alliance.org/authors>
- Summary blurb (enter in submission system): A short text summarizing in a single sentence the study (max. 200 characters)

including spaces). This text is used in conjunction with the titles of papers, hence should be informative and complementary to the title. It should describe the context and significance of the findings for a general readership; it should be written in the present tense and refer to the work in the third person. Author names should not be mentioned.

B. MANUSCRIPT ORGANIZATION AND FORMATTING:

Thank you for your attention to these final processing requirements. Please revise and format the manuscript and upload materials as soon as you are able.

Sincerely,

Reviewer #1 (Comments to the Authors (Required)):

The authors have reasonably addressed my concerns. The experiments regarding the role of H₂O₂ are interesting and add more value to the manuscript.

Reviewer #3 (Comments to the Authors (Required)):

The authors have now added additional experiments to show the selectivity of the ROS on the EE phenotypes in MEFs with mitochondrial dysfunction. The additional experiments adding the H₂O₂ condition, PLK4 overexpression and ROCK inhibition strengthen the manuscript by providing additional evidence that alterations to the centrosome drive the EE clustering. The PLK4 overexpression experiment, in the absence of ROS, strongly argues that the EE phenotypes observed are not due to membrane lipid peroxidation, even if low-grade amounts of peroxidation are happening.

The new added figures, however, need some editing - the Y axis in Sup Fig 3B says "Microtubulin", I believe the authors mean "Microtubules". The same issue is in Sup Fig 5 D and G. In addition, Sup Fig 3C says "Microtubulel" on the Y axis, which should also say "Microtubules".

It would have been helpful to say where exactly in the text the authors implemented changes to the writing, regarding my comments. As I have not been able to open their previous manuscript to physically compare the two line by line, I am unsure which new text they're referring to and cannot properly judge whether this was addressed. In the future, please consider stating the page/line where a change was implemented, or better yet, copy and paste verbatim, the new text directly into the response to reviewers.

Response to the editor/reviewers**Comments from the Editor:**

-Please upload your main manuscript text as an editable doc file.

Done

-Please upload all figure files as individual ones, including the supplementary figure files; all figure legends should only appear in the main manuscript file.

Done

-Please add the X and Bluesky handles of your host institute/organization, as well as your own and/or one of the authors, in our system.

Done

-Please consider changing the title for improved clarity: "Mitochondrial dysfunction impairs early endosome trafficking via microtubule reorganization" or "Oxidative stress impairs early endosome trafficking via microtubule reorganization". Please also ensure the titles in the system and the manuscript file are consistent with each other.

As suggested, we changed the title to "Mitochondrial dysfunction impairs early endosome trafficking via microtubule reorganization"

-The "Data Availability" section should be placed after the Materials & Methods section. Please consult our guidelines at <https://www.life-science-alliance.org/manuscript-prep#format>

This has been added.

-Please add an Author Contributions section to your main manuscript text.

Author contributions have been added

-Please add a Conflict of Interest statement to your main manuscript text.

A conflict of interest statement has been added.

-Please add your main, supplementary figure, and table legends to the main manuscript text after the references section.

Done

-The images in Figure 5B, panel "GO", and 4D, panel "AA" appear identical. Please verify these are correct.

There was indeed a mislabeling of the processed images used to make the figures. Figures 4 and 5 have now been updated to show the proper conditions. Thanks for catching this one.

Reviewer #3

The new added figures, however, need some editing - the Y axis in Sup Fig 3B says "Microtubulin", I believe the authors mean "Microtubules". The same issue is in Sup Fig 5 D and G. In addition, Sup Fig 3C says "Microtubulel" on the Y axis, which should also say "Microtubules".

These issues have now been fixed

It would have been helpful to say where exactly in the text the authors implemented changes to the writing, regarding my comments. As I have not been able to open their previous manuscript to physically compare the two line by line, I am unsure which new text they're referring to and cannot properly judge whether this was addressed. In the future, please consider stating the page/line where a change was implemented, or better yet, copy and paste verbatim, the new text directly into the response to reviewers.

The reviewer is right that it would have been clearer. Sorry for that.

October 2, 2025

RE: Life Science Alliance Manuscript #LSA-2024-03020-TRR

Dr. Marc Germain
Université du Québec à Trois-Rivières
Département de Biologie Médicale
3539 LP
3351, boul. des Forges, C.P. 500
Trois-Rivières, Quebec G9A 5H7
Canada

Dear Dr. Germain,

Thank you for submitting your Research Article entitled "Mitochondrial Dysfunction Alters Early Endosomes Trafficking via Microtubule Reorganization". It is a pleasure to let you know that your manuscript is now accepted for publication in Life Science Alliance. Congratulations on this interesting work.

DISTRIBUTION OF MATERIALS:

Again, congratulations on a very nice paper. I hope you found the review process to be constructive and are pleased with how the manuscript was handled editorially. We look forward to future exciting submissions from your lab.

Sincerely,
